# Intracellular accumulation of amyloid-ß is a marker of selective neuronal vulnerability in Alzheimer's disease

Alessia Caramello [1,2,3], Nurun Fancy [1], Clotilde Tournerie[1], Maxine Eklund[1], Vicky Chau[1], Emily Adair [1], Marianna Papageorgopoulou[1], Nanet Willumsen [1], Johanna S. Jackson [1], John Hardy[2] & Paul M. Matthews [1,4] ✉

Defining how amyloid-β and pTau together lead to neurodegeneration is fundamental to understanding Alzheimer's disease (AD). We used imaging mass cytometry to identify neocortical neuronal subtypes lost with AD in *post-mortem* brain middle temporal gyri from non-diseased and AD donors. Here we showed that L5,6 RORB+FOXP2+ and L3,5,6 GAD1+FOXP2+ neurons, which accumulate amyloid-β intracellularly from early Braak stages, are selectively vulnerable to degeneration in AD, while L3 RORB+GPC5+ neurons, which accumulate pTau but not amyloid-β, are not lost even at late Braak stages. We discovered spatial associations between activated microglia and these vulnerable neurons and found that vulnerable RORB+FOXP2+ neuronal transcriptomes are enriched selectively for pathways involved in inflammation and glycosylation and, with progression to AD, also protein degradation. Our results suggest that the accumulation of intraneuronal amyloid-β, which is associated with glial inflammatory pathology, may contribute to the initiation of degeneration of these vulnerable neurons.

Selective degeneration and loss is observed in a subset of vulnerable neurons at early stages of Alzheimer's disease (AD)[1]. Previous immunohistological studies suggested these include subtypes of excitatory (L2 RELN+ in the entorhinal cortex [EC][2] and L3-5 SMI32+ in the prefrontal cortex [PFC][3]) and inhibitory neurons (SST+ and Calretinin+ in the piriform cortex[4]), while PVALB+ and CALB1+ inhibitory neurons appear resilient to degeneration[5]. Recently, single-cell or nuclei transcriptomic studies identified selective loss in other neuronal subpopulations, such as GAD1+, LHX6+, and NPY+ inhibitory neurons in the PFC[6] and RORB+ and CDH9+ excitatory neurons in the EC[7,8]. However, while single-cell transcriptomics can identify neuronal subpopulations with high resolution, it is subject to multiple confounds for quantitation of cells[9] and lacks information regarding their spatial localisation, particularly in

relation to reactive glia and pathological proteins. The question is important as identifying vulnerable neurons and the local interactions responsible for their selective loss could lead to rational designs of cell-type-targeted diagnostics and novel approaches to treatment.

Neuronal vulnerability has often been associated with intraneuronal accumulation of hyper-phosphorylated Tau (pTau), which forms neurofibrillary tangles (NFT), and can lead to synaptic disfunction, toxicity and neuronal death[10–12]. However, in both the human PFC[6] and AD mouse models[13], selective NFT accumulation appears to be distinct from vulnerability to death, which predominantly occurs in neurons without NFT. Other work has suggested that accumulation of pTau may not impair neuronal function, at least initially[14], and may even enhance resilience to apoptosis[15,16].

[1]UK Dementia Research Institute Centre at Imperial College London and Department of Brain Sciences, 728 Sir Michael Uren Research Hub, 86 Wood Ln, London W12 0BZ, UK. [2]UK Dementia Research Institute Centre at University College London, Department of Neurodegenerative Disease, Wing 1.2 Cruciform Building, Gower Street, London WC1E 6BT, UK. [3]Laboratory of Stem Cell Biology and Developmental Genetics, The Francis Crick Institute, 1 Midland Road, London NW1 1AT, UK. [4]The Rosalind Franklin Institute, Harwell Science and Innovation Campus, Fermi Way, Didcot, Oxon OX11 0QS, UK. ✉e-mail: p.matthews@imperial.ac.uk

An alternative hypothesis relates the accumulation of intraneuronal amyloid-β peptides with selective vulnerability. Aβ–42 is recognised to accumulate intra-neuronally in AD[17]. This accumulation is associated with impaired synaptic functions[18,19] and neurotoxicity[20]. Intra-cellular Aβ–42 (intraAβ) was found to accumulate in RELN[+] EC vulnerable neurons in both rats and humans[21] and may arise with impairment of neuronal autophagy–lysosomal pathways[22,23] implicated by GWAS in susceptibility to AD[24–26]. The decrease in relative numbers of neurons with intraAβ with disease progression supports the hypothesis it may be involved with early neuronal death[27]. Neurodegeneration secondary to intraAβ accumulation could even initiates plaque formation[22,28]. However, the specific neuronal subtypes selectively accumulating intraAβ and their relationship to selective neuronal loss in AD is not known.

However, neurodegeneration may not be cell autonomous. For example, reactive microglia are observed in early AD[29,30] near intraAβ[+] neurons[31]. The high expression of AD risk genes in microglia suggests they play in the initiation of AD[32,33] (and see refs. 34,35). Data regarding the co-localisations of glia with vulnerable neurons is needed.

In this study, we sought to identify the intrinsic cell characteristics, local Aβ and pTau pathology, and glia associated with selective neuronal vulnerability in AD. We also explored selective neuronal vulnerability in tissue from donors heterozygotic for *TREM2* AD risk alleles (*R62H* or *R47H*) expressed in microglia. To do this, we used highly multiplexed imaging mass cytometry (IMC) to map specific excitatory and inhibitory neuronal sub-types, glial cells, and Aβ and pTau pathology in *post-mortem* middle temporal gyri (MTG) samples from non-diseased control (Braak 0-II), early (Braak III-IV) and late (Braak V-VI) AD samples. Our results identified selective loss and early intraAβ for L2,3,5,6 RORB[+]FOXP2[+] and L3,5,6 GAD1[+]FOXP2[+] neuronal populations with AD. Transcriptomes of the vulnerable RORB[+]FOXP2[+] neurons were enriched for pathways involved in unfolded protein responses, protein degradation and glycosylation. Our work thus provides evidence for a direct association between intraAβ accumulation and selective loss of specific neuronal subtypes in AD and suggests that this may be explained in part by intrinsic characteristics of the vulnerable neurons.

## Results

### Multiplexed immunohistology identified neuronal subtypes, their cortical organisation and associated glia

We processed *post-mortem* middle temporal gyrus (MTG) sections from 12 non-diseased controls and 31 AD donors (Fig. 1a; Supplementary Table 1) with imaging mass cytometry (IMC) using an optimised panel of 31 antibodies enriched for neuronal markers (Figs. 1b, c; S1, S2; Supplementary Table 2,3). After automated IMC image processing with SIMPLI[36] (Fig. 1d), 237,248 nuclei (1839 ± 277 nuclei/imaged region of interest (ROI); 1271.6 ± 263.9 nuclei/mm$^2$) were identified, of which 198,470 were positive for at least one cell marker. These were clustered and assigned to neuronal or glial subpopulations based on cell-type-specific marker expression in each cluster (Figs. 2a, b; S3).

The relative proportions of neurons that expressed excitatory (68.6 ± 1.8%; including clusters of unclassified neurons) or inhibitory (31.4 ± 1.8%) markers in CtrlCV sections were consistent with previous observations[8,37] (Fig. 2d). The proportion of astrocytes to total glial cells (45.5 ± 5.7%) also was within the expected range, although the relative abundances of oligodendrocytes (26.7 ± 6.5%) and microglia (27.7 ± 3.5%) appeared to be under- and over-represented, respectively, relative to a prior stereological report[38]. This discrepancy is likely due to the limited number of markers used for identifying microglia, which will bias towards overcounting these cells, and the relatively weak IMC signal from OLIG2, which may lead to undercounting of oligodendrocytes. Neuronal subtypes from CtrlCV samples were found to have layer-specific distributions that were generally consistent with those expected (e.g., PCP4[+] neurons in L5[39] [cluster_23]

and CUX2[+] neurons in L2/3 [cluster_13][6]; PVALB[+] neurons L3/5 [cluster_29] and SST[+] L5/6 [cluster_14][40]) (Fig 2e).

### RORB[+] and GAD1[+] neurons account for the majority of neuronal loss with AD

We tested whether distinguishable neuronal subtypes are selectively lost in late-onset Braak 5-6 AD tissue (AlzCV; n = 18) compared to that from non-diseased control Braak 0-2 donors (CtrlCV; n = 6). We found selective reductions in the numbers of five neuronal subtypes with AD (Fig. 3a). Three of these expressed RORB alone or in combination with other subtype-specific markers (RORB[+]MAP2[+] [cluster_5], p < 0.001; RORB[+]FOXP2[+] [cluster_8], p < 0.001; RORB[+] [cluster_11], p = 0.029). These three RORB-expressing clusters accounted for 81.4% of the total neuronal loss observed in AlzCV cases compared to CtrlCV (86.5 ± 146.2 neurons/mm$^2$ lost per AlzCV sample; 70.4 ± 55.4 RORB-expressing neurons/mm$^2$ lost per AlzCV sample). Additional neuronal subtypes showing smaller relative decreases in cells number with AD corresponded to GAD1[+]ADARB1[+] (cluster_9, p = 0.029) and ADARB1[+] (cluster_28, p = 0.044) inhibitory neurons. We conclude that these five neuronal subtypes are selectively vulnerable to cell loss with AD.

We then explored whether vulnerable neurons were lost in specific neocortical layers (Fig. 3b). We found a lower density of RORB[+]FOXP2[+] neurons (cluster_8) with AD predominantly in L3 (p = 0.033), L4 (p = 0.047), and L6 (p = 0.022). GAD1[+]ADARB1[+] neurons (cluster_9) were most abundant in L6 but showed the greatest relative reduction in density with AD in L4 (p = 0.018). Despite not being reduced in total number, fewer GAD1[+]FOXP2[+] neurons (cluster_32) were found in L3-6 with AD (p = 0.003, 0.029, 0.022, and 0.005, respectively). A lower density of RORB[+]GPC5[+] neurons (cluster_10) was found in L1 (p = 0.005) in AlzCV compared to controls. Conversely, we observed a higher density of RORB[+]GPC5[+] neurons in AlzTREM2 cases compared to CtrlCV in L6 (p = 0.033).

Our results thus identified selective loss of both excitatory and inhibitory vulnerable neuronal populations with AD. Most of these were localised in L3, 4, and 6 and expressed the RORB and GAD1 markers previously linked to vulnerable neurons[7,41].

### Neuronal loss is greater in brains from donors carrying TREM2 risk variants

*TREM2* risk variants are associated with increased risk of AD, earlier disease onset, and faster disease progression[34,42,43]. To test whether the expression of *TREM2* risk variants increases neuronal loss and whether these are the same neuronal populations lost in sporadic AD cases, we extended our analyses to include samples from non-diseased control and AD donors heterozygotic for *TREM2 R47H* or *R62H* allelic variants (CtrlTREM2, n = 6; 2/6 carrying the *R47H* allele; AlzTREM2, n = 13; 6/13 carrying *R47H* allele). In AlzTREM2 samples, the same neuronal subtypes were selectively vulnerable in AlzTREM2 sample as in AlzCV, with decreased numbers of RORB[+]MAP2[+] (cluster_5; p = 0.008), RORB[+]FOXP2[+] (cluster_8; p < 0.001), RORB[+] (cluster_11; p = 0.004), GAD1[+]ADARB1[+] (cluster_9; p = 0.007) and ADARB1[+] (cluster_28; p = 0.039) neuronal subtypes relative to CtrlCV (Fig. 3c). Consistent with a more aggressive clinical disease phenotype[34,42,43] compared to AlzCV, AlzTREM2 samples had lower numbers of RORB[+]FOXP2[+] (cluster_8; −36.3 ± 15.8%), RORB[+] (cluster_11; −27.6 ± 15.9%) and ADARB1[+] (cluster_28; −16.1 ± 13.7%) neurons. Additionally, we found fewer GAD1[+]FOXP2[+] neurons (cluster_32; p = 0.045) in AlzTREM2 samples than in CtrlCV. Greater numbers of RORB[+]GPC5[+] neurons (cluster_10; p = 0.036) were found in AlzTREM2 sections than in CtrlTREM2. No significant differences were found between AD samples and controls in the total immunostained areas for the synaptic markers synaptophysin and NTNG2 (Figure S5.a).

We found layer-specific vulnerabilities in AlzTREM2 cases compared to CtrlCV in L5,6 for RORB[+]FOXP2[+] neurons (cluster_8; p = 0.019 for both), L6 for GAD1[+]FOXP2[+] neurons (cluster_32; p = 0.025) and L5 for

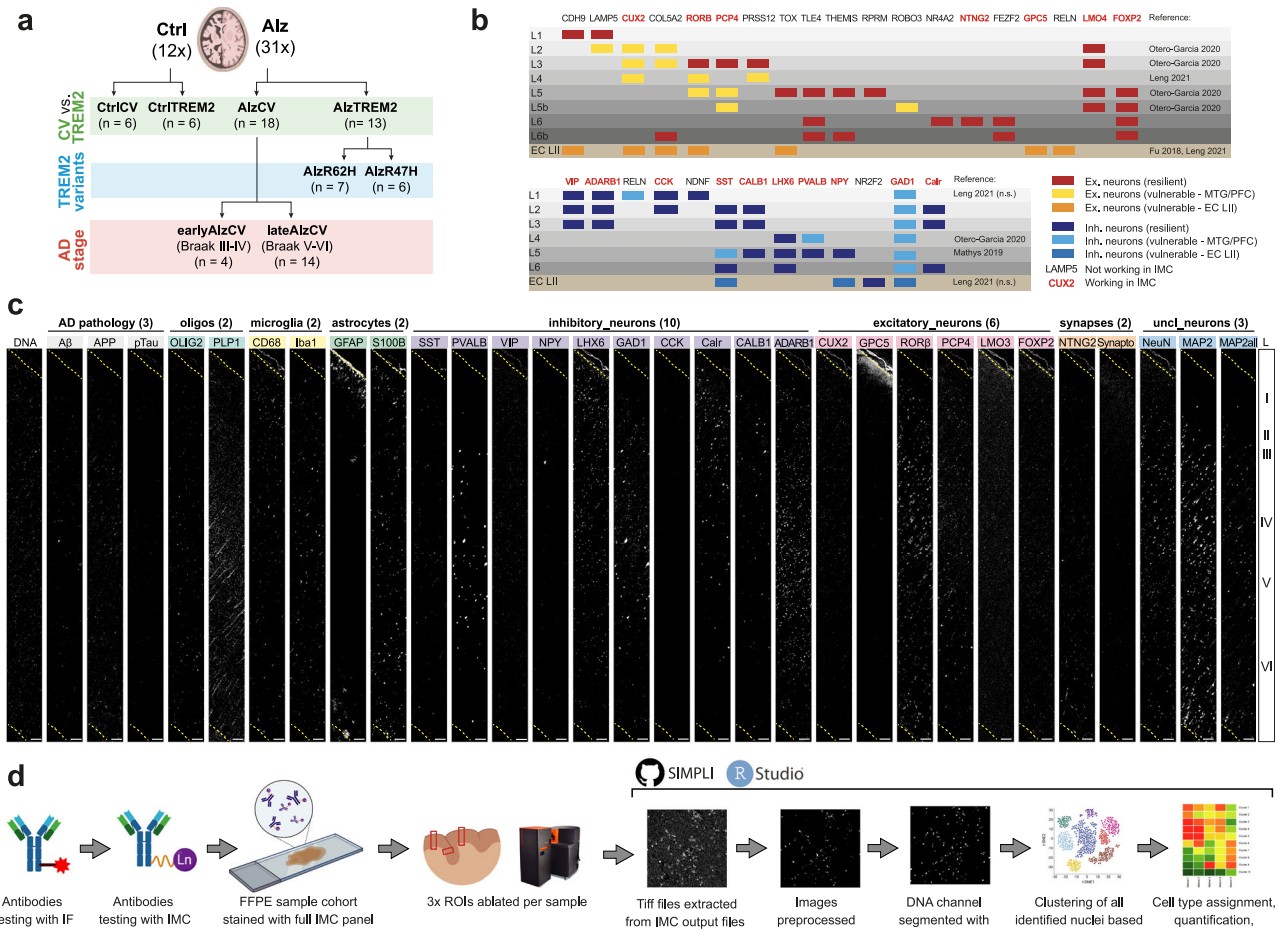

**Fig. 1 | Use of imaging mass cytometry (IMC) for identification of neuronal and glial subtypes in human post-mortem brain. a** FFPE sections of MTG from 12 non-disease controls and 31 AD cases were processed for IMC. Depending on the analysis, samples were divided based on expression of TREM2 common allele (CV) or TREM2 risk variants (*R62H* and *R47H*) and Braak stages. **b** List of candidate neuronal markers tested in developing the IMC antibody panel with their expected cortical distributions in middle temporal gyrus (MTG), prefrontal cortex (PFC), and entorhinal cortex (EC) layer II, previous associations with neuronal vulnerability, and corresponding references. Antibodies included in the final panel are indicated in red. **c** Example of IMC images obtained from one region of interest (ROI) processed with the final antibody panel. Each ROI ablated spans the entire thickness of the neocortex cortex (L1-L6). **d** A cartoon illustrating the full methodological pipeline from antibody testing (first by immunofluorescence microscopy [IF] and then using IMC), staining and ablation of full sample cohort, automated image analysis using SIMPLI (see Methods) to final data analysis in R. Graphics were created in BioRender (Caramello, A. (2025) https://BioRender.com/z22okvn)[103]. Scale bars in **c** represent 100 μm.

GAD1+ neurons (cluster_33; $p = 0.007$) (Fig. 3d; S3c). Compared to CtrlTREM2, AlzTREM2 cases showed lower densities of PCP4+ neurons (clusters_23) in L4 ($p = 0.047$) and GAD1+ neurons (cluster_33) in L1 ($p = 0.037$). A few neuronal populations showed layer-specific increases in cell density in AlzTREM2 cases compared to CtrlTREM2 (L5 CUX2+ neurons [cluster_13; $p = 0.037$]; L3 MAP2+MAP2all+ neurons [cluster_17; $p = 0.044$]; L6 NPY+ neurons [cluster_26; $p = 0.008$]). The previously identified differences between CtrlCV and AlzCV samples in layer-specific neuronal densities (Fig. 3b) still showed the same trend despite the loss of statistical significance with multiple testing corrections.

Together, these results suggest that neuronal loss is greater in AD patients carrying *TREM2* risk variants and involves the same subtypes that are selectively vulnerable in AD with the common *TREM2* allele. We interpret the relatively greater density of a few neuronal subpopulations in AD as evidence for their relative resilience with overall loss of neuropil.

## The *TREM2 R47H* risk variant is associated with greater neuronal loss with AD

The relative AD risk conferred by the *TREM2 R47H* variant is higher than that of the *R62H* variant[44,45]. To test whether higher AD risk is associated with greater relative neuronal loss in AD, we re-analysed our dataset after splitting the data from AlzTREM2 samples based on their *TREM2* genotypes (AlzR62H [$n = 7$] and AlzR47H [$n = 6$]). Among the five neuronal subpopulations showing evidence for selective loss in AlzTREM2 cases, four were significantly reduced only in AlzR47H cases compared to CtrlCV: RORB+MAP2+ (cluster_5; $p = 0.038$), GAD1+ADARB1+ (cluster_9; $p < 0.001$), RORB+ (cluster_11; $p = 0.014$) and ADARB1+ (cluster_28; $p = 0.041$) subtypes (Fig. 3e). RORB+FOXP2+ neurons (cluster_8) were significantly reduced in both AlzR62H and AlzR47H samples compared to CtrlCV ($p < 0.001$ for both).

Layer-specific neuronal loss relative to CtrlCV was found in AlzR47H cases in L6 GAD1+FOXP2+ neurons (cluster_32) ($p = 0.042$; Fig. 3f). Conversely, AlzR62H sections showed selective loss of L2 and L6 RORB+FOXP2+ neurons (cluster_8) compared to both AlzCV and CtrlCV donors ($p = 0.041$ for both) and L6 PCP4+ neurons (cluster_23) compared to AlzCV donors ($p = 0.035$). L5 GAD1+ neurons (cluster_33) were lost in both AlzR47H ($p = 0.043$) and AlzR62H ($p = 0.017$) cases compared to CtrlCV. Together, these results confirm that the *TREM2 R47H* heterozygotes showed the greatest neuronal loss and widest range of neurons showing loss with AD.

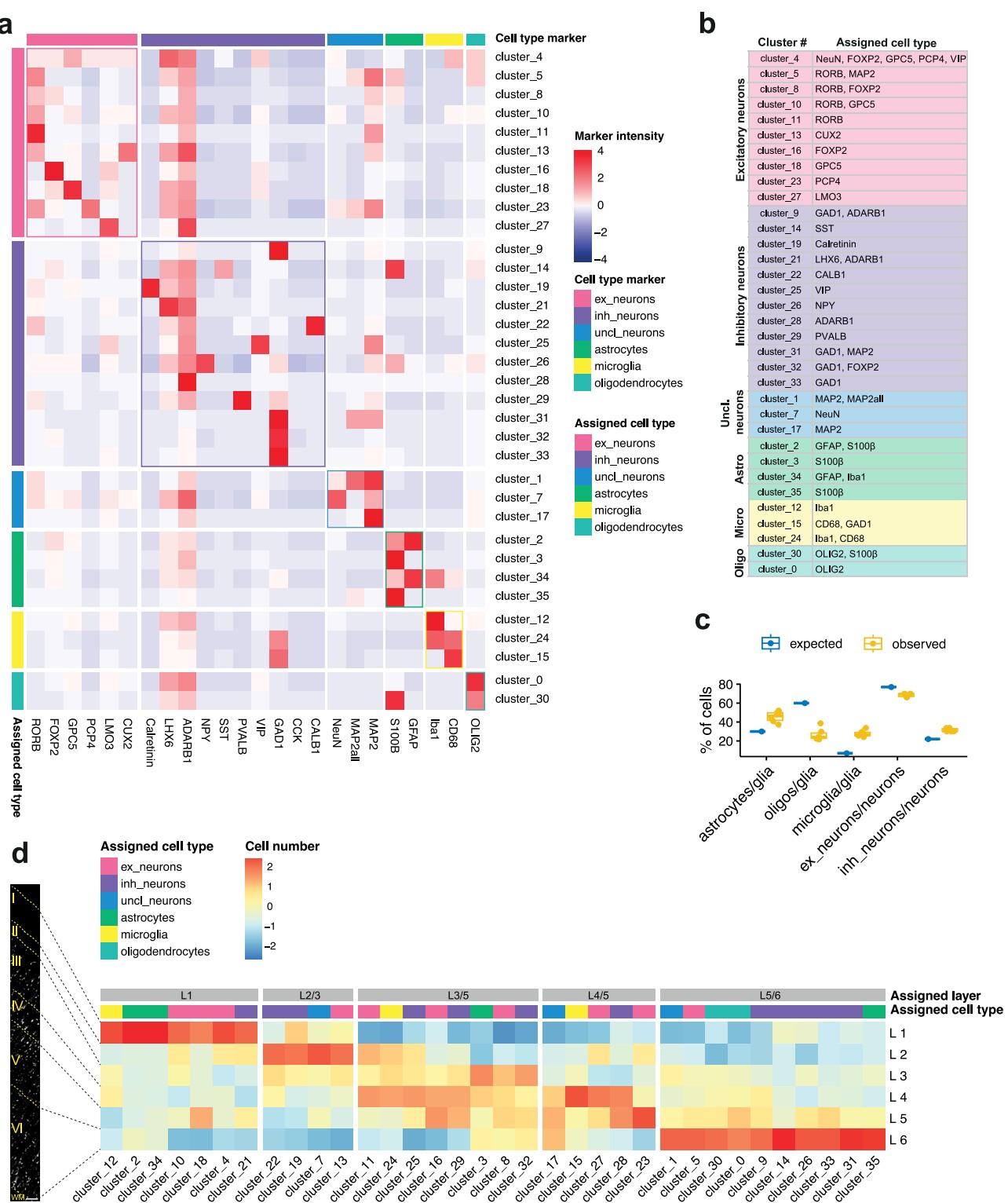

**Fig. 2 | Characteristic markers and cell type assignments for clusters of nuclei detected in IMC. a** Heatmap showing relative mean intensities of marker expression (columns) in each identified cluster (rows). Markers were grouped based on their cell-type specificities (see top of plot, "cell type marker"). Their relative intensities in each cluster were used to assign clusters to cell types (see left of plot, "assigned cell type"). **b** Final assignment of neuronal and glial cell types to each cluster, with main markers expressed by each cluster indicated on the right. **c** Observed and expected proportions of astrocytes, oligodendrocytes, and microglia relative to the total glia population (left) and of excitatory (which include unclassified neurons) or inhibitory neurons to the total neuronal population (right)

in the CtrlCV samples (sections from n = 6 brains). **d** Distributions of cells from each cluster within the 6 cortical layers (CtrlCV samples only), normalised by total numbers of nuclei per layer. Cell type and layer assigned to clusters are indicated in the coloured "Assigned cell type" and grey "Assigned layer" boxes above, respectively. Quantification was performed on three ROIs acquired from a single section of each sample and pooled together. Boxplots in (**c**) show median (middle line), interquartile range (box), and variability outside of first and third quartiles (lines extending from box). Source data are provided as a Source Data file. Scale bars in (**d**) represent 100 μm.

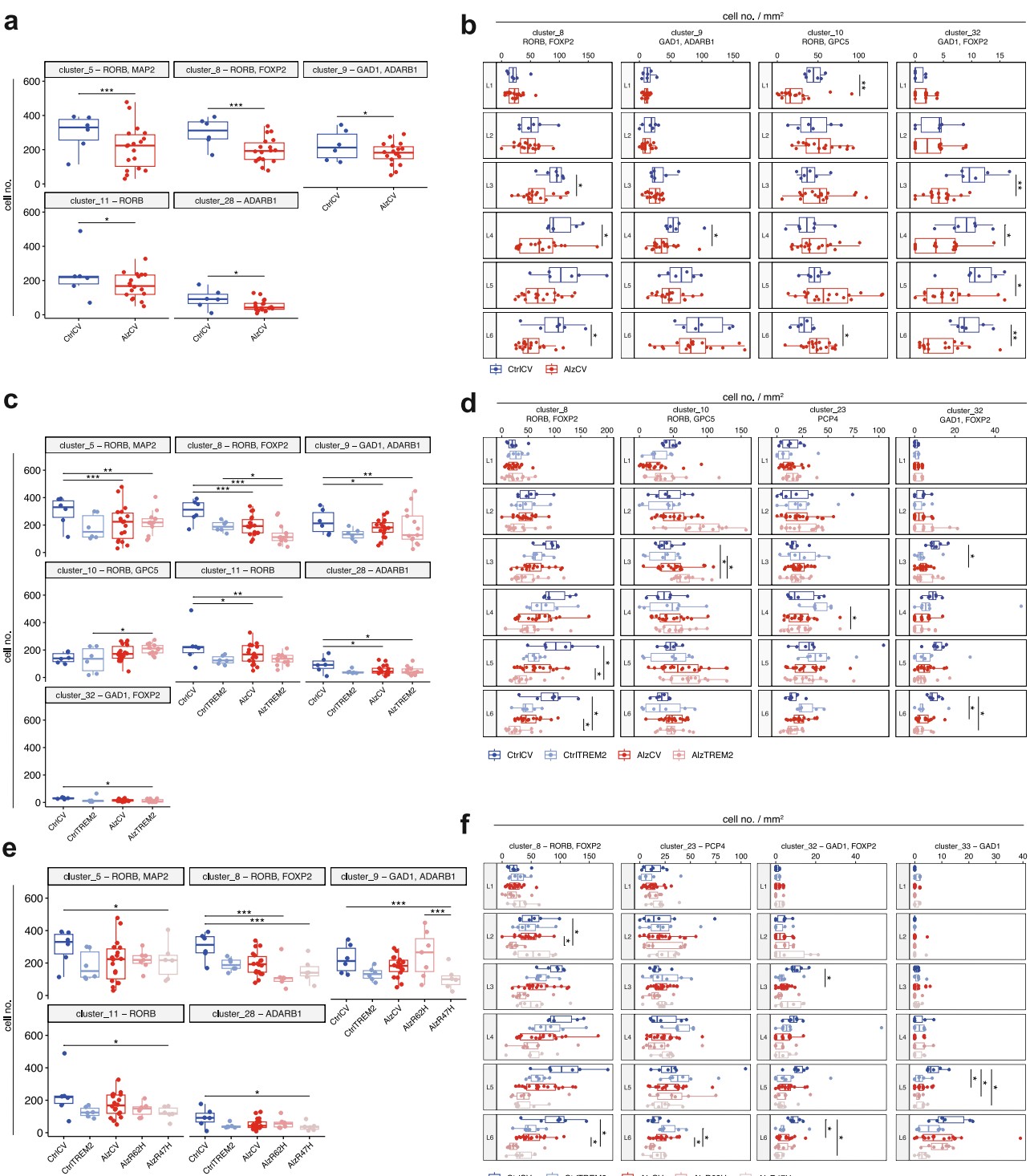

**Fig. 3 | Neuronal subpopulations and cortical layers selectively affected in AD.** Average cell numbers per cluster per sample, comparing either controls and AD samples from donors heterozygotic for the common TREM2 allele (**a**; CtrlCV, sections from n = 6 brains; AlzCV, *n* = 18) or controls and AD samples homozygotic for the TREM2 common allele and those heterozygotic for R62H or R47H risk variants combined (c; CtrlCV, *n* = 6; CtrlTREM2, *n* = 6; AlzCV, *n* = 18; AlzTREM2, *n* = 13) or separately (e; CtrlCV, *n* = 6; CtrlTREM2, *n* = 6; AlzCV, *n* = 18; AlzR62H, *n* = 7; AlzR47H, *n* = 6). Only clusters with significant differences in sizes between sample groups are shown. The density of cells (cell number/mm²) in each cortical layer (L1-6) from all clusters showing layer-specific vulnerabilities (distribution of all neuronal clusters shown in Figure S5.b), comparing either controls and AD samples carrying the common TREM2 allele (**c**; CtrlCV, *n* = 6; AlzCV, *n* = 18) or controls and AD samples homozygotic for TREM2 common allele and those heterozygotic for

R62H or R47H risk variants combined (**d**; CtrlCV, *n* = 6; CtrlTREM2, *n* = 6; AlzCV, *n* = 18; AlzTREM2, *n* = 13) or considered separately (**d**; CtrlCV, *n* = 6; CtrlTREM2, *n* = 6; AlzCV, *n* = 18; AlzR62H, *n* = 7; AlzR47H, *n* = 6). Quantification was performed on three ROIs acquired from a single section of each sample and pooled together before performing statistical analyses between groups. Statistical significance was calculated with Dirichlet regression (**a,c,e**), two-sided Wilcoxon signed-rank test (**b**), or either ANOVA and two-sided Tukey tests or Kruskal–Wallis and two-sided Wilcoxon signed-rank test, depending on whether groups showed normal or non-normal distributions, respectively (**d**, **f**). Boxplots show median (middle line), interquartile range (box), and variability outside of the first and third quartile (lines extending from the box). P values are indicated as: non-significant, ns, *p* > 0.05; *$p \leq 0.05$; **$p \leq 0.01$; ***$p \leq 0.001$; ****$p \leq 0.0001$. Source data are provided as a Source Data file.

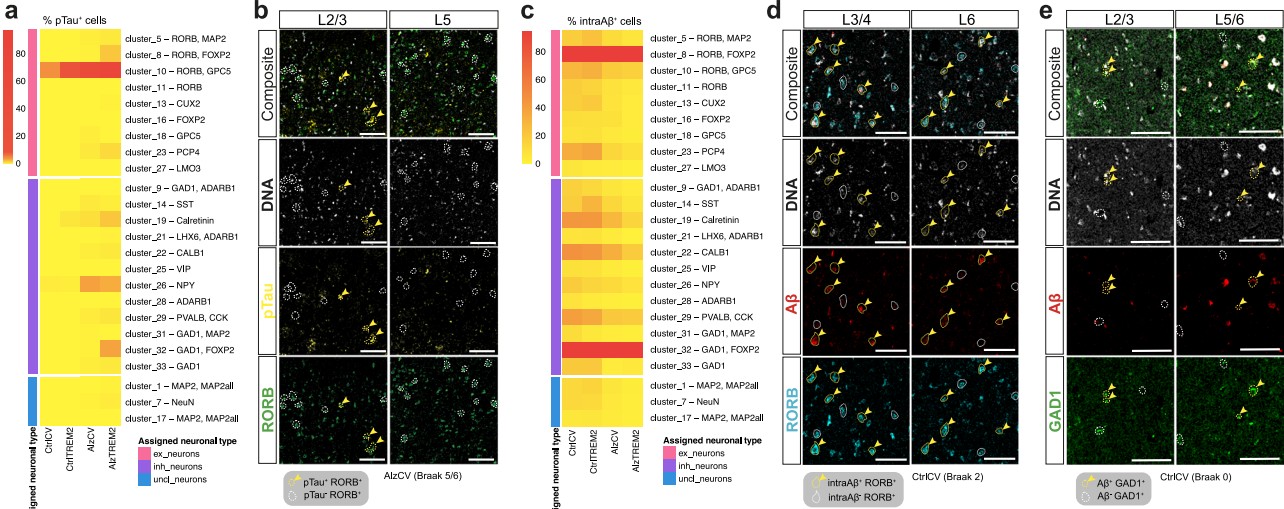

**Fig. 4 | Intracellular accumulation of AT8⁺ pTau and 4G8⁺ Aβ (intraAβ) in neuronal subtypes.** Proportions of pTau⁺ (**a**) and intraAβ⁺ (**c**) cells in each neuronal subtype cluster separately for CtrlCV (sections from $n = 6$ brains), CtrlTREM2 ($n = 6$), AlzCV ($n = 18$) and AlzTREM2 ($n = 13$) samples. Quantification was performed on three ROIs acquired from a single section of each sample and pooled together. Coloured boxes on the left highlight groups of neuronal types (excitatory, inhibitory and unclassified neurons) to which clusters were assigned to ("assigned neuronal type"). IMC images showing co-localisations of: RORB⁺ and AT8⁺ pTau in a AlzCV (Braak 5/6) tissue section (**b**); RORB⁺ neurons and 4G8⁺ Aβ in a CtrlCV (Braak 2) section (**d**); GAD1⁺ neurons and 4G8⁺ Aβ in a CtrlCV (Braak 0) section (**e**). Double-positive neurons (RORB⁺AT8⁺ in **b**; RORB⁺4G8⁺ in **d**; GAD1⁺4G8⁺ in **e**; yellow arrowheads) are found in the cortical layers shown in Fig. 3b, d, f. Source data are provided as a Source Data file. Scale bars in (b,d,e) represent 100 µm.

## Intracellular accumulation of AT8⁺ pTau is not a major determinant of neuronal loss

To explore the relationships between pTau accumulation and neuronal vulnerability, AlzCV samples were split into early- (Braak 3-4) and late-AD (Braak 5-6) cases. As expected, total AT8⁺ pTau immunostaining (Fig. S6a) increased in later Braak stages (AlzCV Braak 5-6 vs. AlzCV Braak 3-4, $p = 0.006$) and showed a trend to be greater in AlzTREM2 cases compared to AlzCV Braak 5-6 ($p = 0.07$; Fig. S6.b), but was similar between donors carrying the two *TREM2* risk variants (Fig. S6d). Intraneuronal AT8⁺ pTau signal (pTau⁺ neurons; Fig. S6f) was more frequently found in neurons of AlzCV Braak stages 5-6 and AlzTREM2 donors compared to CtrlCV ($p = 0.001$ for both), CtrlTREM2 ($p = 0.001$ for both) and AlzCV Braak stages 3-4 donors ($p = 0.009$ for both) (Fig. S6h). The proportion of total pTau⁺ neurons was similar between AlzTREM2 and AlzCV Braak stages 5-6 donors and between donors carrying different *TREM2* risk variants (Fig. S6.j). These results confirm that, as expected, pTau expression increases in the MTG with the progression of AD pathology.

To test whether AT8⁺ pTau preferentially accumulates in specific neuronal subpopulations, we quantified the proportions of pTau⁺ neurons in each neuronal subtype cluster. The proportion of pTau⁺ cells was greatest in RORB⁺GPC5⁺ neurons (cluster_10 with $68.52 \pm 12.58\%$ pTau⁺ neurons) in AlzTREM2 cases (Fig. 4a). These RORB⁺GPC5⁺pTau⁺ neurons predominantly localised to L2/3 (Fig. 4b). A small proportion of RORB⁺GPC5⁺pTau⁺ neurons were also found in non-diseased control donor tissue sections ($4.07 \pm 3.1\%$). Lower proportions of pTau containing neurons were found in NPY⁺ (cluster_26; $3.38 \pm 3.94\%$), GAD1⁺FOXP2⁺ (cluster_32; $3.22 \pm 9.19\%$), RORB⁺FOXP2⁺ (cluster_8; $1.86 \pm 2.51\%$), calretinin⁺ (cluster_19; $1.88 \pm 2.2\%$) and PCP4⁺ (cluster_23; $1.08 \pm 2.42\%$) neurons in AlzCV or AlzTREM2 cases. Our results thus show that pTau⁺ primarily accumulate in RORB⁺GPC5⁺ neurons and can be found from as early as Braak stages 0-2, suggesting they might be uniquely susceptible to NFT formation. However, because the density of RORB⁺GPC5⁺ neurons does not decrease with AD, susceptibility to AT8⁺ pTau accumulation alone is not a major determinant of cell death in AD.

## Intracellular accumulation of β-amyloid is a marker of neuronal vulnerability to cell death in AD

We then tested for associations between the total 4G8⁺ immunostained area and intracellular Aβ accumulation with neuronal loss. As expected, the total 4G8⁺ immunostained Aβ area (Fig. S6a) was greater in AlzCV Braak stages 5-6 and AlzTREM2 donors compared to both CtrlCV ($p = 0.003$ for both) and CtrlTREM2 ($p = 0.011$, $0.003$, respectively) non-diseased donors (Fig. S6c). AlzCV Braak stages 3-4 donors also showed significantly lower total 4G8⁺ Aβ area compared to both AlzCV Braak stages 5-6 and AlzTREM2 donors ($p = 0.037$, $0.011$, respectively). Total Aβ deposition was similar between AlzCV Braak stage 5-6 and AlzTREM2 donors and donors carrying either of the two *TREM2* risk variants (Fig. S6e). Conversely, the proportion of all detected neurons showing intracellular 4G8⁺ Aβ immunostaining (intraAβ; Fig. S6g) was highest in CtrlCV ($25.3 \pm 5.22\%$) and CtrlTREM2 ($25.29 \pm 8.24\%$) and significantly lower in AlzCV Braak stage 5-6 ($16.66 \pm 3.71\%$; relative to either CtrlCV [$p = 0.011$] or CtrlTREM2 [$p = 0.013$]) and in AlzTREM2 ($14.05 \pm 3.55\%$; relative to either CtrlCV [$p < 0.001$] or CtrlTREM2 [$p < 0.001$]) donor sections (Fig. S6.i). We did not find significant differences in proportions of intraAβ⁺ neurons between AlzTREM2 and AlzCV Braak stages 5-6 donor sections or between sections from donors heterozygotic for the two *TREM2* risk variants (Fig. S6.k). These results suggest either that intraAβ accumulation begins at an early Braak stage in what otherwise might be considered as "healthy" or non-diseased tissue and then decreases at more advanced Braak stages, or that neurons accumulating intraAβ are particularly vulnerable to cell death with AD progression.

To determine whether specific neuronal subtypes selectively accumulate intraAβ, we quantified the proportions of intraAβ⁺ neurons in each neuronal subtype cluster that we defined. Virtually all RORB⁺FOXP2⁺ (cluster_8; on average $99.2 \pm 1.8\%$) and GAD1⁺FOXP2⁺ (cluster_32; on average $99.7 \pm 1.9\%$) neurons were positive for intraAβ regardless of AD stage (Fig. 4c). Lower proportions of intraAβ⁺ cells were found in CtrlCV or CtrlTREM2 donor sections in CALB1⁺ (cluster_22; $46.26 \pm 5.9\%$), calretinin⁺ (cluster_19; $43.97 \pm 5.81\%$), PVALB⁺CCK⁺ (cluster_29; $40.48 \pm 11.39\%$) and PCP4⁺ (cluster_23;

33.55 ± 11.11%) neuronal subtypes. The proportions of intraAβ+ neurons were lower at higher Braak stages for all but RORB+FOXP2+ (cluster_8) and GAD1+FOXP2+ (cluster_32) neurons. IntraAβ+RORB+ and intraAβ+GAD1+ neurons were found predominantly in middle (L2,3,4) and lower (L5,6) cortical layers of CtrlCV tissue (Fig. 4d, e). Layers 2/3 and 5/6 are those in which loss of RORB+FOXP2+ (cluster_8) and GAD1+FOXP2+ (cluster_32) neurons is more significant (Fig. 3b, d, f).

The 4G8 antibody has been used previously for intraAβ studies[22], but can also bind to full-length amyloid precursor protein (APP) and its cleaved products, such as the amyloid precursor protein β-C-terminal fragment (βCTF)[46]. To confirm assignment of the amyloid binding signal to Aβ, we repeated studies using the Aβ42-specific antibody MOAB-2[47] using triple immunofluorescence with anti-GAD1 or anti-RORB together with the pan-neuronal marker MAP2. We found that GAD1+ and RORB+ neurons co-localised with MOAB-2 immunostaining (representative images from CtrlCV [Braak 2], CtrlTREM2 [Braak 0], AlzCV [Braak 6] and AlzTREM2 [Braak 6] samples shown in Fig. S7a, d) which was found intracellularly (Fig. S7b, c, e, f). Thus, in this context, intracellular 4G8 signal can be interpreted as evidence of intracellular Aβ accumulation.

To summarise, our results show firstly that intraAβ and pTau preferentially accumulate in different neuronal subtypes. Secondly, intraAβ accumulation begins from the earliest Braak stages particularly in RORB+FOXP2+ and GAD1+FOXP2+ neurons. The progressively lower relative numbers of these neuronal subtypes with intraAβ in tissues sections from brains at higher Braak stages is consistent with their selective loss. From this, we conclude that intraAβ is a marker of vulnerability to AD-associated neuronal loss.

### Activated microglia and astrocytes increase with AD, particularly in patients heterozygotic for the TREM2 R62H variant

Both microglia and astrocytes can phagocytose injured or metabolically stressed neurons and may make major direct contributions to neuronal loss with AD[48–50]. Overall, numbers of Iba1+CD68+ microglia (cluster_24) were similar in non-diseased control and AD donor tissues (Fig. S8a). However, CD68+GAD1+ cell numbers were greater in donors carrying TREM2 risk variants compared to either CtrlCV ($p = 0.045$) or CtrlTREM2 ($p = 0.034$) tissues (Fig. 5a). The relatively greater number of CD68+GAD1+ cells was most significantly associated with the TREM2 R62H variant relative to CtrlCV ($p = 0.007$) or AlzCV ($p = 0.016$) tissue sections (Fig. 5b). To confirm the co-localisation of these markers and the identity of the CD68+GAD1+ cells, we performed triple marker (Iba1+, CD68+, and GAD1+) immunofluorescence on sections from some of the same brains. Orthogonal projection of these images identified Iba1+CD68+ cells with microglial morphologies and confirmed co-localisation of intracellular GAD1 in both non-diseased control and AlzTREM2 sections (Fig. 5c). These results suggest CD68+GAD1+ (cluster_15) cells, which are more abundant in TREM2 risk variant tissue sections, define microglia that have phagocytosed GAD1+ inhibitory neurons or their synapses.

Reactive GFAP+S100B+ astrocytes[51] (cluster_2) were more abundant in AlzTREM2 cases compared to CtrlCV ($p = 0.023$) and AlzCV ($p = 0.063$) (Fig. S8.b), particularly in sections from donors heterozygotic for the TREM2 R62H variant relative to AlzCV ($p = 0.016$) (Fig. S8.c). We did not find significant differences in the number of OLIG2+ oligodendrocytes (cluster_0) or in PLP1+ myelin area (Fig. S8.d,e). Together, these results suggest that later AD stages are characterised by more active microglial phagocytosis and greater astrocytic activation.

### Activated microglia, but not astrocytes, are spatially associated with vulnerable neurons and amyloid plaques

To determine whether reactive microglia and astrocytes contribute to neuronal vulnerability in AD by physically interacting with vulnerable neurons, we performed spatial interaction analyses. Among all

samples, we found that CD68+GAD1+ microglia interacted with RORB+GPC5+ (cluster_10), RORB+FOXP2+ (cluster_8), and LMO3+ (cluster_27) excitatory (Fig. 5d) and PVALB+CCK+ (cluster_29), CALB1+ (cluster_22), and GAD1+FOXP2+ (cluster_32) inhibitory neurons (Fig. 5e). Therefore, half of the neuronal clusters spatially associated with CD68+GAD1+ describe neuronal populations accumulating pathological proteins. Conversely, among the astrocytic clusters, only S100B+GFAP- astrocytes (cluster_3, cluster_35) showed preferential spatial interactions with neuronal clusters. These were found to be associated with 9 neuronal clusters, only 2 of which included vulnerable neurons (RORB+MAP2+ [cluster_5] and GAD1+ADARB1+ [cluster_9]), neither of which showed evidence for accumulation of either Aβ or pTau (Fig. S8f, g). Together, these data suggest that activated microglia may preferentially cluster around vulnerable neurons and neurons accumulating pTau, while homoeostatic astrocytes have a weaker spatial association with vulnerable neurons and are associated more frequently with neurons more resilient to loss.

We then tested whether reactive glia and amyloid plaques were localised preferentially in the cortical layers showing greatest associations with vulnerable neurons. We found a higher CD68+GAD1+ microglia density in L3-6 of AlzTREM2 donors compared to CtrlTREM2 ($p = 0.013, 0.045, 0.008, 0.058$ for each of L3-6, respectively; Fig. 5f). A similar but non-significant trend was also found in L3 of AlzCV sections compared to CtrlCV ($p = 0.082$). By contrast, GFAP+S100B+ astrocytes (cluster_2) were mostly localised in L1 and did not show layer-specific differences with AD (Fig. S8.h). Therefore, activated microglia, but not astrocytes, show a layer-specific distribution similar to that of the vulnerable neurons (Fig. 3b, d, f). We then characterised the distribution of amyloid plaque density across the neocortical layers. We found that more than 70% of total plaque area was in L3 (29.4%), L5 (20.8%), and L6 (20.6%) of the AlzTREM2 and AlzCV samples (Fig. 5g; quantified as pixel area of 4G8+ amyloid plaque masks Fig. S8i). Amyloid plaques are thus also relatively more abundant in L3,5 and 6 where vulnerable neurons are located.

Finally, to assess glial associations with plaques, we quantified the proportions of CD68+GAD1+ microglia (cluster_15) and GFAP+S100B+ astrocytes (cluster_2) within and 50 μm around amyloid plaques. Compared to CtrlCV, both AlzCV and AlzTREM2 cases showed higher proportions of CD68+GAD1+ microglia and GFAP+S100B+ astrocytes around (microglia, $p = 0.012, 0.008$, respectively; astrocytes, $p = 0.018$ for both) and within plaques (microglia, $p = 0.054$ for both; astrocytes, $p = 0.041, 0.075$, respectively) (Fig. S8.j–m). However, within plaques of AlzTREM2 cases, proportions of microglia ($4.53 ± 4.7\%$) were higher than astrocytes ($0.62 ± 0.84\%$; t-test $p = 0.011$). Our spatial distribution and interaction analyses thus provide evidence that microglia, but not astrocytes, are associated with both vulnerable neurons and amyloid plaques in AlzTREM2 and AlzCV donor sections.

### Exploration of mechanisms of neuronal vulnerability using single-nucleus transcriptomics

To explore pathways and molecular mechanisms associated with vulnerability of the neuronal subtypes identified, we analysed a single-nucleus RNA sequencing (snRNAseq) dataset (described in an earlier preprint from our group)[52] generated from cryo-preserved blocks from homologous MTG regions of the same brains studied above, along with 10 additional MTG samples to enhance study power (3 CtrlCV, 1 CtrlTREM2, 1 AlzCV, and 5 AlzTREM2) (Supplementary Table 4). Nuclei were clustered to identify transcriptionally distinguishable sub-populations of excitatory and inhibitory neurons (Fig. 6a). These then were matched to neuronal subtypes identified with IMC based on scores for similarities of cluster sizes and subtype marker expression (defined as read counts for snRNAseq cluster markers [Fig. S9.a] and relative intensities of immunomarker expression for IMC) (Supplementary Data file 1). Based on the highest matching scores (Fig. S9b), we found eleven transcriptionally defined

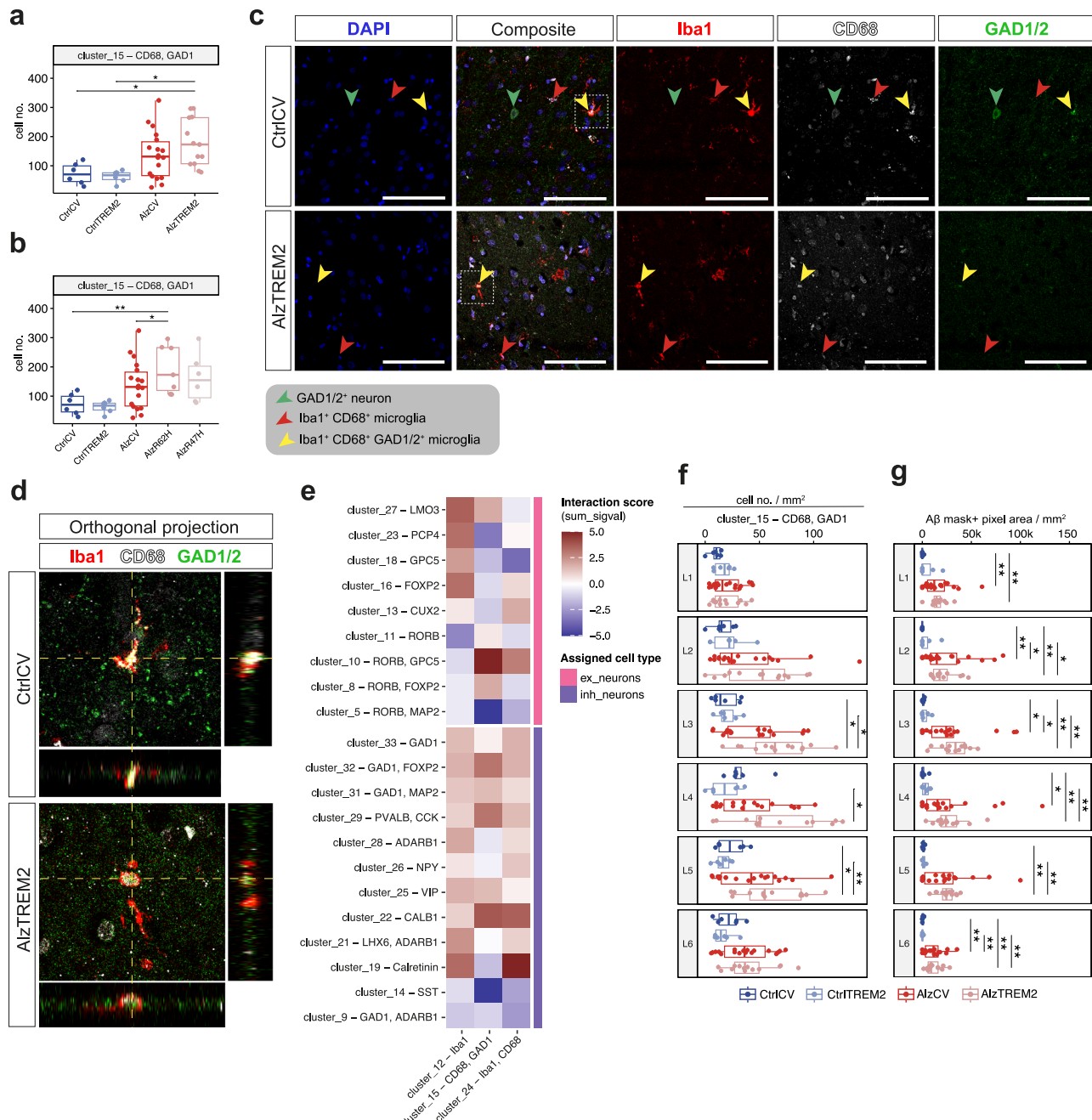

**Fig. 5 | Reactive microglia are spatially associated with vulnerable neurons and plaques.** Average number of CD68⁺GAD1⁺ cells (cluster_15) per sample group in sections from non-disease control or AD donors homozygotic for the TREM2 common allele or heterozygotic for TREM2 R62H or R47H risk variants with the latter analysed together (**a**; CtrlCV, *n* = 6; CtrlTREM2, *n* = 6; AlzCV, *n* = 18; Alz-TREM2, *n* = 13) or separately (**b**; CtrlCV, *n* = 6; CtrlTREM2, *n* = 6; AlzCV, *n* = 18; AlzR62H, *n* = 7; AlzR47H, *n* = 6). **c** Triple immunostaining for Iba1 (red), GAD1 (green), CD68 (grey) and DAPI (blue) in CtrlCV and AlzTREM2 donor sections with showing triple positive Iba1⁺CD68⁺GAD1⁺cells (yellow arrowhead). **d** Orthogonal projection of the region indicated by the white dashed square in (c). **e** Cell-cell interaction analyses between microglia and inhibitory or excitatory neuronal clusters from all samples analysed were performed with the buildSpatialGraph function. The calculated interaction score "sum_sigval" indicates the relative interactions between (>0) or relative proximity avoidance (<0) of the selected cell types. **f** Densities of CD68⁺GAD1⁺ cells by cortical layer (cluster_15) in CtrlCV (*n* = 6),

CtrlTREM2 (*n* = 6), AlzCV (n = 18) and AlzTREM2 (*n* = 13) samples. **g** Layer-specific density of Aβ⁺ mask (shown in Fig. S8.i) generated from the 4G8 Aβ channel to identify plaques and their total areas within the sections of CtrlCV (n = 6), CtrlTREM2 (*n* = 6), AlzCV (*n* = 18), and AlzTREM2 (*n* = 13) samples. Quantification was performed on three ROIs acquired from a single section of each sample and pooled together before performing statistical analyses between groups. Statistical significance estimates were calculated with Dirichlet regression (**a**, **b**) or either ANOVA and two-sided Tukey tests or Kruskal–Wallis and two-sided Wilcoxon signed-rank test depending on whether groups showed normal or non-normal distributions, respectively (**f**, **g**). Boxplots show median (middle line), interquartile range (box) and variability outside of the first and third quartiles (lines extending from box). *P* values are indicated as: non-significant, ns, *p* > 0.05; *\**p* ≤ 0.05; *\*\**p* ≤ 0.01; *\*\*\**p* ≤ 0.001; *\*\*\*\**p* ≤ 0.0001. Source data are provided as a Source Data file. Scale bars in (**c**) represent 100 μm.

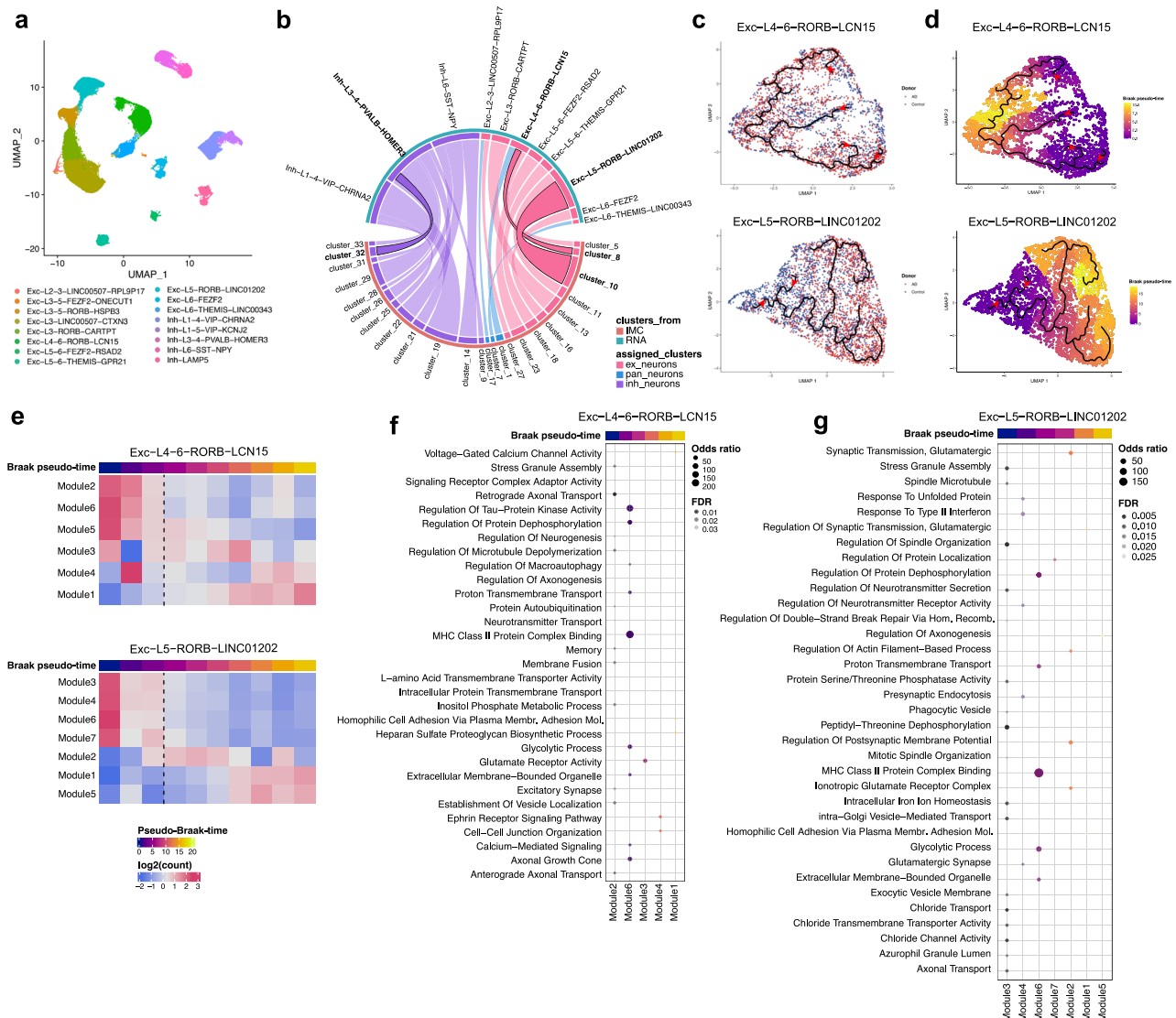

**Fig. 6 | Pathways enriched in Braak pseudo-time trajectory analyses of IMC-snRNAseq matched vulnerable neuronal clusters. a** UMAP representation of clustered MTG-derived neuronal subpopulations from snRNAseq dataset. **b** Matched neuronal clusters from IMC (orange line below) and snRNAseq (green line above) experiments. Colours of the connecting lines identify neuronal types (excitatory, inhibitory, and unclassified neurons) to which clusters were assigned to ("assigned neuronal subtype"). Relative widths of the connecting lines describe their relative match scores (a measure of confidence in the correspondence between the direct transcriptomic and immunohistological cluster associations with wider lines corresponding to higher matching scores). Summary clinical disease (left) and Braak (right) pseudo-time trajectories for the Exc−L4−6−RORB−LCN15 and Exc−L5−RORB−LINC01202 clusters (black line in **c,d** with trajectory directions indicated by the red arrowhead). **e** Heatmap describing the relative expression of gene pathways enriched at successive stages (modules) of Braak pseudo-time for the Exc−L4−6−RORB−LCN15 (upper) and Exc−L5−RORB−LINC01202 (lower) clusters. Pathways enriched in modules corresponding to the successive consecutive Braak pseudo-time stages identified for Exc−L4−6−RORB−LCN15 (**f**) and Exc−L5−RORB − LINC01202 (**g**) cluster trajectory analyses. Only pathways with FDR < 0.05, odds ratio > 8 and overlapping genes ≥ 3 were analysed.

neuronal clusters that were well-matched to neuronal subtype clusters identified by IMC (Fig. 6b; Supplementary Table 5). These included clusters defined by IMC as neuronal subtypes accumulating intraAβ and selectively vulnerable to loss with AD (RORB⁺FOXP2⁺ [cluster_8] and GAD1⁺FOXP2⁺ [cluster_32] clusters matching with Exc−L4−6−RORB−LCN15 and Inh−L3−4−PVALB−HOMER3, respectively) and another neuronal subtype accumulating pTau but without evidence for selective loss with AD (RORB⁺GPC5⁺ [cluster_10] matching with Exc−L5−RORB−LINC01202). Despite matching, we did not find good correspondences between relative reductions in nuclear counts in AD brains compared to non-diseased controls in snRNAseq-defined clusters matching with IMC-defined vulnerable neurons (Fig. S9c). This may be expected given the multiple factors other than cell abundance that determine nuclear counts in snRNAseq cluster[53,54].

## Transcriptomic identification of gene expression pathways that distinguish vulnerable neuronal subpopulations

To explore molecular mechanisms associated with neuronal subtype vulnerability, we explored gene expression pathways relatively enriched in each neuronal subtypes using snRNAseq (Fig. S10.a; Supplementary Data file 2). We first tested for enrichment of neuronal subtype clusters for expression of pathways potentially relevant to AD.

In the Exc−L4−6−RORB−LCN15 cluster, corresponding to the vulnerable RORB⁺FOXP2⁺ neurons accumulating intraAβ, the "Response to IL6" inflammatory pathway was enriched uniquely

relative to other neuronal clusters, as were those for glycosylation and sialylation processes (e.g., "Sialyltransferase Activity" and "Glyco-sphingolipid Biosynthetic Process"), dysfunction of which have previously been reported for AD[55], as well as the "Heparan Sulfate Proteoglycan Metabolic Process", including heparan sulfate 3-O-sulphotransferase 2 (HS3ST2), which modulates brain amyloid-β clearance and aggregation[56] and "3′,5′-cyclic AMP phosphodiesterase activity". We also found significant enrichment in several other pathways, shared with other neuronal clusters, e.g., related to the regulation of calcium channel activity (e.g., "Regulation Of Voltage−Gated Calcium Channel Activity") and synapse assembly and transmission (e.g., "Syntaxin-1 Binding" and "Synaptic Vesicle Cycle"), the latter including the synuclein alpha (SNCA) transcript encoding for α-synuclein which can contribute to the pathophysiology of AD[57], and Piccolo Presynaptic Cytomatrix Protein (PCLO), which encodes a presynaptic cytoskeletal protein, variants in which have been linked to increased AD risk[58].

Fewer pathways were identified in the Exc−L5 − RORB − LINC01202 cluster, corresponding to the RORB⁺GPC5⁺ neuronal subtype accumulating pTau. This cluster showed enrichment for GABAergic synaptic transmission-related pathways (e.g., "Synaptic Transmission, GABAergic" and "GABA−A Receptor Complex"), glycosylation processes (e.g., "Protein C−linked Glycosylation Via 2′−alpha−mannosyl−L−tryptophan"), and heparan sulfate-related pathways (e.g., "[Heparan Sulfate] −Glucosamine 3−Sulfotransferase 1 Activity").

Pathways enriched in the Inh−L3−4−PVALB−HOMER3 cluster, corresponding to GAD1⁺FOXP2⁺ vulnerable neurons accumulating intraAβ, were mostly related to ion channel activity (e.g., "Voltage-Gated Monoatomic Ion Channel Activity" and "Store-Operated Calcium Channel Activity") and glutamatergic synapses (e.g., "Postsynaptic Density Membrane"). The latter pathway included the Glutamate Ionotropic Receptor NMDA Type Subunit 2B (GRIN2B), expression of which previously was reported to be altered with AD[59]. The "Heparan Sulfate Sulfotransferase Activity" pathway was also enriched in this cluster with expression of the Heparan Sulfate-Glucosamine 3-Sulfotransferase 1 (HS3ST1) gene encoding a protein which binds to ApoE[60] and has been linked to AD risk through GWAS[61].

In conclusion, this analysis identified pathway enrichments that could contribute to neuronal vulnerability. The IL6 inflammatory response pathway was enriched in the vulnerable Exc−L4−6−RORB − LCN15 cluster, which, along with the Inh−L3−4−PVALB−HOMER3 cluster, was also enriched in pathways related to calcium or ion channels. Glycosylation and heparan sulfate-related pathways were enriched in neuronal clusters that accumulate intraAβ or pTau.

## AD-relevant pathways are differentially expressed in vulnerable neuronal subpopulations with disease progression

We extended our exploration of gene expression patterns related to neuronal subtype vulnerability by performing pathway analyses of differentially expressed genes with a categorical contrast for diagnosis (AD or non-diseased control) or by regression for relative levels of tissue 4G8⁺ β-amyloid or PHF1⁺ pTau immunostaining[52] (Fig. S10b; Supplementary Data file 3).

The Exc−L4−6−RORB−LCN15 cluster, corresponding to the vulnerable RORB⁺FOXP2⁺ neurons, showed decreased expression of the "proton transmembrane transport" pathway, which includes several components of the Vacuolar ATPase (V-ATPase) Complex (e.g., ATP6V1A, ATP6V0B, ATP6V0E1) in the categorial contrast for diagnosis. V-ATPase regulates lysosomal acidification for protein degradation, and dysfunction has been associated with AD previously[62]. The "regulation of autophagy of mitochondrion" pathway was also downregulated in the same clusters. This pathway includes Glycogen Synthase Kinase 3 Alpha (GSK3A), which is involved in the pathological production of amyloid-β peptides in AD[63], and the voltage-dependent

anion channel 1 (VDAC1), which can potentiate amyloid-β to cause mitochondrial dysfunction and neuronal death[64,65], as well as ubiquitin-proteasome system genes such as FBXO7 and USP36. However, as these pathways were also downregulated within the apparently resilient Exc-L6-FEZF2 and Exc-L6-THEMIS-LINC00343 (for V-ATPase) and Exc-L5-6-FEZF2-RSAD2 (for autophagy and ubiquitin) clusters, they cannot be independent determinants of vulnerability.

The Inh−L3−4−PVALB−HOMER3 cluster, corresponding to vulnerable GAD1⁺FOXP2⁺ neurons accumulating intraAβ, showed downregulation of pathways related to synaptic function ("regulation of long-term synaptic plasticity", "synaptic vesicle", and "from presynaptic endocytic zone" with the categorical contrast for diagnosis). We found differential expression in pathways similarly affected in other neural clusters. Pathways upregulated included those related to glycosylation (e.g., "protein O−linked glycosylation" and "other types of O-glycan biosynthesis") and synthesis of glycoproteins and glycolipids (e.g., "oligosaccharide metabolic process", "glycosphingolipid metabolic process" and "protein O-linked mannosylation") for all three differential expression analyses. As mentioned above, previous reports have highlighted dysfunction of both protein glycosylation[55] and sphingolipid synthesis in AD[66]. Similarly, several pathways related to cell-cell adhesion (e.g., "regulation of cell-cell adhesion" and "heterophilic cell-cell adhesion via plasma membrane cell adhesion molecules") were also consistently upregulated in this cluster with all three analyses.

These results thus extend the description of pathways associated with relative vulnerability for neurons that accumulate intraAβ. The pathways highlight processes with established roles in AD pathology, such as protein degradation, glycosylation, and the synthesis of glycoproteins.

## Relating transcriptional pathway expression in vulnerable neurons to Braak stage progression

A complementary approach to exploring mechanisms that may be associated with vulnerability is to test whether pathways identified by differential expression analyses are relatively enriched at the early or late stages of AD. For this, we performed trajectory analyses within our three vulnerable neuronal transcriptional clusters, in addition to another neuronal cluster that did not show evidence of loss even in late disease stages (Exc-L6-THEMIS-LINC00343, matching to the IMC-defined NeuN⁺ cluster_7; Supplementary Table 5). A pseudo-time trajectory related to disease progression from Braak 0-2 (early stage) to Braak 5-6 (late stage) was developed for each cluster (Figs. 6c, d; S11a, b) and used to define transcriptional modules representing early, mid and late stages of the disease (Figs. 6e; S11c). We then performed cluster-specific pathway enrichment analyses for each module across the Braak pseudo-time course (Fig. 6f, g; S11d, e; Supplementary Data File 4).

There was some evidence for differences in enrichment or differences in disease stage for enrichments of transcriptional modules of vulnerable neurons along Braak pseudo-time trajectories. For example, cell stress and death pathways were associated with the earliest Braak pseudo-time stage in the Exc−L4−6−RORB−LCN15, Exc−L5−RORB−LINC01202 and Inh−L3−4−PVALB−HOMER3 clusters accumulating either intraAβ or pTau ("Stress Granule Assembly" in Modules M2 and M3 in the Exc−L4−6−RORB−LCN15 and Exc−L5−RORB−LINC01202 clusters, respectively, Fig. 6f, g; "Regulation Of Neuron Death" in M4 in Inh−L3−4−PVALB−HOMER3 cluster, Fig. S11d). The latter pathway was also enriched in the resilient L6-THEMIS-LINC00343 neuronal cluster (M2; Fig. S11e), but in mid stages.

Enrichments in pathways related to autophagy and protein degradation (e.g., "protein auto-ubiquitination" in M2, "regulation of macroautophagy" in M6; Fig. 6.f) were associated with early disease stage in cluster Exc−L4−6−RORB − LCN15, corresponding to

RORB[+]FOXP2[+] vulnerable neurons (Fig. 6g). Glycosylation pathways (e.g., heparan sulfate proteoglycan biosynthetic process") were enriched in mid- or late stages in M1 and M3 of Exc−L4−6−RORB−LCN15 and Inh−L3−4−PVALB−HOMER3 clusters, respectively (Figs. 6.f and S11d). The related "glycosaminoglycan biosynthesis" pathway was enriched at an early disease stage in the L6-THEMIS-LINC00343 cluster (M1, Fig. S11e).

Exc-L4-6-RORB-LCN15 is the only neuronal cluster showing enrichment for the "Regulation Of Macroautophagy" and "Protein Autoubiquitination" at the early or mid-stages of disease (M6 and M2, respectively). In other cases, selective enrichment or enrichment pathways by disease stage were not apparent for pathways of possible pathological significance. Chloride channel-related pathways were enriched at the early or mid-stages of disease in both Exc−L5−RORB−LINC01202 and L6-THEMIS-LINC00343 clusters, (e.g., "chloride channel activity" in M3 and M2, respectively; Figs. 6g, S11e). Chloride channels regulate neuronal excitability, and their homoeostasis is altered in AD[67,68]. Glycolytic pathways (e.g., "glycolysis/gluconeogenesis" and "glycolytic process" pathways) were similarly enriched in all clusters at early and mid-stages of the Braak pseudo-time (M6 in Exc−L4−6−RORB−LCN15, M6 in Exc−L5−RORB−LINC01202, M4 in Inh−L3−4−PVALB−HOMER3, and M2 in L6-THEMIS-LINC00343; Fig. 6f, g and Figure S11d, e).

Our Braak pseudo-time analysis thus describes neuronal pathway enrichments with AD progression. Stress response, protein degradation and neuronal death appear to be relatively enriched at early disease stages in neurons vulnerable to degeneration.

## Discussion

We combined IMC and snRNAseq to identify the MTG neocortical neuronal subtypes preferentially lost with AD and explore mechanisms underlying their vulnerability. We found evidence for selective and cortical-layer specific loss of seven neuronal subpopulations, the majority of which expressed RORB or GAD1, both previously identified markers of AD vulnerable neurons. The same subtypes of neurons showed trends for greater selective neuronal loss with AD in donors heterozygotic for the *TREM2 R47H* or *R62H* variants associated with earlier onset and more rapid progression of disease. The vulnerable RORB[+]FOXP2[+] and GAD1[+]FOXP2[+] neuronal subtypes accumulated intraAβ early in the progression of pathology. By contrast, increased intracellular pTau accumulation at higher Braak stages was associated with RORB[+]GPC5[+] neuronal subtypes, which did not show significant neuronal loss. We also provide evidence that activated CD68[+] microglia were preferentially spatially associated with the vulnerable neuronal subtypes accumulating intraAβ. By contrast, reactive S100B[+]GFAP[+] astrocytes were spatially associated with neurons that did not decrease in numbers even at late Braak stages. Transcriptomic analyses of snRNAseq-IMC matched vulnerable neuron clusters allowed identification of AD-relevant pathways potentially contributing to selective neuronal susceptibility to AD pathology. Our study thus provides evidence that a high level of intraAβ accumulation is an early pathological marker of the neuronal subtypes selectively lost with the progression of AD in the MTG, and that susceptibility to pTau accumulation can be decoupled from susceptibility to this cell loss. Our results add to the growing evidence for inflammation-associated impairments in autophagy[69] and protein degradation[70] as mechanisms in the initiation of cortical neurodegeneration in AD. They also highlight the importance of further understanding roles for glycosylation processes[55] and chloride channels[68] in AD-related neurodegeneration.

The majority of vulnerable neurons markers we identified had been associated previously with selective neuronal loss in other brain regions (RORB[+], GAD1[+] and GPC5[+] in EC[7,8] and PFC[6], respectively, and see ref. 71). This suggests common mechanisms underlying neuronal vulnerability that are shared across brain regions. We found that ADARB1[+] neurons also appeared relatively vulnerable, as reported previously[72]. We did not find significant reductions in other neuronal subpopulations previously identified as vulnerable in AD, such as CUX2[+] and SST[+] neurons[6,7,52]. We also identified a relative layer-specific increase in cell density of a few neuronal subpopulations (e.g., RORB[+]GPC5[+], CUX2[+], and NPY[+] neurons). We interpret this as most likely reflecting relative resilience to neurodegeneration in the context of the wider reduction in neuropil and cortical thickness with AD[73].

Our results provided evidence for relatively selective accumulation of intraAβ in neurons vulnerable to loss with the progression of AD, and that fewer neurons with intraAβ were found with higher Braak stages. A similar finding was reported with higher pathology load in Aβ-expressing transgenic mouse models[27]. This suggests either that intraAβ is cleared from neurons as pathology progresses or that neurons accumulating Aβ are lost due to cell death. We interpret the association between vulnerable neuronal subpopulations and high intraAβ load as most consistent with the latter hypothesis. However, it is important to note that we did not find that all vulnerable neuronal subtypes were associated with intraAβ accumulation (e.g., ADARB1[+] neurons) and found that neuronal populations showing low levels of intraAβ accumulation were not vulnerable to loss with AD progression, e.g., calretinin[+], CALB1[+], and PVALB[+] inhibitory neurons.

Consistent with previous reports in humans and mouse models of AD, we found a neuronal subtype-specific susceptibility to accumulation of pTau (RORB[+]GPC5[+] neurons), which, however, appeared to be decoupled from vulnerability to cell loss[6,13,74]. The increased numbers of pTau[+] cells in a few neuronal subpopulations at later stages of the disease, in fact, suggest that it may distinguish relative neuronal resilience to death[15,16].

By including sections from brain donors heterozygotic for *TREM2* risk variants, we were able to test both whether the risk variants associated with earlier onset disease and rapid progression led to greater neurodegeneration and whether the same neuronal subtypes were selectively vulnerable as with the *TREM2* common allele. We found that AlzTREM2 cases were characterized by greater relative neuronal loss than AlzCV. *TREM2* risk variants thus potentiate degeneration of neurons. We also found a trend for a lower proportion of intraAβ[+] neurons in AlzTREM2 cases compared to AlzCV, consistent with their relative association with degenerating neurons. CtrlTREM2 samples had neuronal numbers and densities in the low range of those associated with CtrlCV. Whether differences could arise from developmental expression of TREM2 variant proteins deserves further investigation[75].

We recognise limitations of our work. First, the immunohistology using IMC was limited to a single imaging plane. This could reduce both accuracy and precision for quantitative assessments of cell numbers, but is unlikely to have a major impact on relative neuronal counts in the neocortex because of the comparable sizes of the neuronal nuclei by which they are detected. Second, neuronal subtypes defined using mRNA sequencing and immunohistology were able to be matched only approximately, given differences in the numbers of markers used to distinguish the neuronal subtypes by the two approaches. A more extensive set of antibodies could better describe relationships between glial sub-states and neuronal subtype variability. The nature of the intraAβ species accumulating in vulnerable neurons also needs to be explored further using a broader range of antibodies specific for different forms of Aβ, e.g., by including the Aβ−42 specific antibody MOAB-2 in future IMC studies. Finally, defining neuronal loss in terms of differences in numbers of neurons in a cluster assumes that neurons maintain a consistent phenotype with disease progression, which is difficult to confirm. Nevertheless, the congruence of our results with previous studies suggests the overall robustness of our approach.

In conclusion, although intraAβ presence and toxicity in neurons has been reported previously, as well as its accumulation in neurons of

AD vulnerable brain regions[76], our study establishes a direct association between selective intraAβ accumulation and neuronal loss in AD. Testing for causal links between early intraAβ and the degeneration of vulnerable neuronal subtypes may provide insights regarding questions fundamental to the so-called "amyloid hypothesis" of AD[77]. Our results raise the question of whether mechanisms responsible for accumulation of intraAβ could explain the fundamental link between amyloid-β and pTau in the genesis and progression of AD[78].

## Methods

### Tissue Samples
This study was carried out in accordance with the Regional Ethics Committee and Imperial College Use of Human Tissue guidelines. Cases were selected based first on neuropathological diagnosis (non-disease control [NDC] or AD) from UK brain banks (London Neuro-degenerative Diseases Brain Bank [King's College London], Newcastle Brain Tissue Resource, Queen's Square Brain Bank [University College London], Manchester Brain Bank, Oxford Brain Bank and South West Dementia Brain Bank [University of Bristol]. We excluded cases with clinical or pathological evidence for small vessel disease, stroke, cerebral amyloid angiopathy, diabetes, Lewy body pathology (TDP-43), or other neurological diseases. Where the information was available, cases were selected with a *post mortem* delay of less than 49 hours. The final cohort (Supplementary Table 1,4; Fig. 1a) was formed of 12 non-diseased controls (Braak 0-II) and 31 early (Braak III-IV; 4 samples) and late (Braak V-VI; 27 samples) AD human *post-mortem* FFPE middle temporal gyrus (MTG) samples. Of these, 4 controls and 7 late AD donors carried the AD high-risk *R62H* TREM2 variant, while 2 controls and 6 late AD donors carried the *R47H* variant. For antibody optimisation, one prefrontal cortex (PFC) control sample (age range: 81-85; sex: male; *post mortem* delay: 48 hours) was obtained as an FFPE block from Parkinson's UK Brain Bank (Imperial College London). All the brain banks used have generic national research ethics committee approval to function as research tissue banks and therefore did not require additional ethics panel approvals for use by UK researchers.

### Designing and testing of the neuronal antibody panel for IMC
An initial list of candidate markers for excitatory and inhibitory neurons and synapses, including vulnerable neurons, was identified by screening previous literature that used single nuclei RNAseq transcriptomic analyses, RNAscope, and antibody-based stainings on fresh frozen or formalin-fixed paraffin-embedded (FFPE) *post mortem* human brain tissue[37,79–82] to determine neuronal subtype- and cortical layer-specific markers. The initial list was refined to prioritise markers characterised in multiple prior publications and to include at least one marker per cortical layer or known neuronal subpopulation and markers previously identified for vulnerable neurons[1,7,83]. To develop a panel applicable in future applications to different human brain regions, markers shared by neurons in the entorhinal (EC), middle temporal gyrus (MTG), and prefrontal cortex (PFC) were prioritised. Antibodies commercially available in a carrier-free solution were prioritised to reduce the need for further cleaning before use as antibody carriers can inhibit IMC metal isotope conjugation reactions.

In preliminary work, we explored binding of antibodies for 34 neuronal subpopulations first by immunofluorescence (IF) together with known pan-neuronal or synaptic markers (MAP2, MAP2all, NeuN and synaptophysin) and then with IMC after trial conjugations to metal isotope chelates (Fig. 1.b; Supplementary Table 2). We found that six excitatory (CUX2, GPC5, RORB, PCP4, LMO3, FOXP2) and ten inhibitory (SST, PVALB, VIP, NPY, LHX6, GAD1, CCK, CR, CALB1, ADARB1) neuronal markers provided comparable immunostaining with both IF and IMC (markers in red in Fig. 1.b; Figure S1). The final IMC antibody panel used for this study included these latter 16 neuronal markers in conjunction with three pan-neuronal markers (NeuN, MAP2 and MAP2all, the latter covering also short isoforms of MAP2)[84], two

synaptic markers (synaptophysin, NTNG2), six glial marker for astrocytes (S100B and GFAP), microglia (Iba1 and CD68), oligodendrocytes (OLIG2), myelin (PLP1), three markers of AD-associated proteins (amyloid-β, pTau and APP) and an iridium nuclear marker (Fig. 1c, Figure S2). The OLIG2 marker gave a relatively less strong and more variable IMC signal than for other markers. Imaging regions of interest across the full thickness of the cortex (pia to leukocortical boundary) allowed their spatial relationships and organisation by cortical layer to be mapped (Fig. 1c).

### Immunofluorescence staining and confocal acquisition
The PFC control sample FFPE block used for antibody optimisation was sectioned at 8 μm at the microtome and placed over super frost glass slides. Both locally sectioned and sections provided by brain banks (also 8 μm thick) were baked overnight at 60 °C to allow FFPE sections to adhere to the slide. Sections then were deparaffinised and rehydrated with consecutive 5 minutes incubations in Histo-Clear II (2x; National Diagnostics), 100% ethanol (2x), 90% ethanol and 70% ethanol and finally washed in water for 5 minutes. Antigen retrieval was performed by incubating slides in EDTA pH 9.0 for 20 minutes in a steam chamber, then cooled down for 10 minutes in ice and washed once in water and once in PBS for 5 minute each. After applying a hydrophobic barrier around the tissue, slides were incubated in blocking buffer (10% donkey serum in PBS 0.3% Triton X-100) for an hour at room temperature inside a humid chamber. Primary antibodies were diluted (as indicated in Table 2) in blocking buffer and applied on slides overnight (for 1 or 2 nights) at 4 °C. Slides where then washed 3x in PBS before applying secondary antibodies (Alexa Fluor Cross Absorbed made in Donkey) and 4',6-diamindino-2-phenylindole (DAPI) diluted 1:500 in blocking buffer and incubated for 1 hour at room temperature in a dark humid chamber. Finally, slides were washed twice with water for 5 minutes each, incubated with Sudan Black (1% in 70% ethanol) for 10 minutes in a dark humid chamber, then washed with running tap water for 10 minute and air dried. Cover slips were mounted with ProLong™ Diamond Antifade Mountant (Invitrogen). Stained sections were acquired at the Leica TCS SP8 confocal microscope at 20x.

### Antibody conjugation, staining and acquisition of imaging mass cytometry (IMC)
Primary antibodies that were not suspended in a carrier-free solution were first purified using antibody purification kits (Abcam mouse antibody, Protein A or G kits) depending on the antibody type and species and following manufacturer instructions (Supplementary Table 2). Carrier-free and purified antibodies then were conjugated to lanthanide metals using the Maxpar X8 kit (Fluidigm, Standard Bio-Tools, CA, USA) according to the manufacturer instructions.

One FFPE section per sample was stained with the full conjugated antibody panel (Supplementary Table 3) following the same protocol as for immunofluorescence. After overnight incubation with metal-tagged antibodies and an iridium (Ir191/193) intercalator (Standard BioTools, CA, USA) to identify nuclei, slides were washed with water 3×10 minutes each and air dried for 20 minutes. IMC was performed using a Hyperion Tissue Imager coupled to a Helios mass cytometer (Fluidigm, Standard BioTools, CA, USA). The instrument was first tuned using the manufacturer's 3-Element Full Coverage Tuning Slide before the slides were loaded into the device. Three ROIs were ablated from the same FFPE section for each sample at a laser frequency of 200 Hz with 1μm resolution. Each ROI was spanning the full cortical depth from L1 to L6 (grey matter – representative ROI in Fig. 1.B) adding up to a total of 6 hours of acquisition per sample (~2 h per ROI, corresponding to $1.2 \pm 0.15 \text{ mm}^2$ of area ablated per ROI) keeping the total area acquired per sample equal (total $3.62 \pm 0.11 \text{ mm}^2$ ablated area per sample). Data was stored as .txt files that were used for subsequent processing (Fig. 1d).

## Automated IMC image analysis

The SIMPLI (v1.1.0) pipeline[36] was used for automated image processing and analysis (Fig. 1d). SIMPLI first transforms.txt files for each ROI to TIFF images and performs normalisation and other pre-processing using CellProfiler[85]. The threshold smoothing scale, correction factor, lower and upper bounds, and manual threshold in the CellProfiler pipeline for image pre-processing was adjusted for each channel to remove background and keep specific signal only. These were adapted in pilot work to yield a total number of positive cells per channel that was comparable to that from manual counting. All images were processed with the same CellProfiler pipeline. Single-nuclei segmentation was then performed within SIMPLI based on the intercalator (191Ir/193Ir) channel using StarDist[86], with the "2D_versatile_fluo" model and a probability threshold of 0.05. Single-nuclei channels intensity was used by SIMPLI for masking all identified nuclei and identify all cells that expressed at least one of the markers used (the "all_cells" subset), apart from those for amyloid-β, pTau and PLP1. To identify neuronal and non-neuronal subtypes, the "all_cells" subset underwent unsupervised clustering and representation in UMAP space with Seurat (resolution 0.9) using again all neuronal and non-neuronal markers but excluding PLP1 marker signals (Figure S3.a).

## Assignment of cell types to clusters

Markers expressed within each cluster (Fig. 2a; Figure S3.b) were used to assign clusters to neuronal/glial subtypes (Fig. 2b): 10 clusters were assigned to excitatory neurons, 12 to inhibitory neurons, 3 to unclassified neurons, 2 to synaptic markers, 4 to astrocytes, 3 to microglia and 2 to oligodendroglia.

The largest cluster was marked by OLIG2+ (cluster_0), which, together with a smaller OLIG2+S100B+ cluster (cluster_30), reflected the high abundance of oligodendroglial cells in the cortex. Two main clusters representing reactive GFAP+S100B+ (cluster_2) and non-reactive GFAP-S100B+ (cluster_30) astrocytes, including two smaller similar clusters (cluster_34, cluster_35) were found. Less cells in smaller clusters were assigned to Iba1+ microglia that showed either a CD68+ (cluster_24) or CD68- (cluster_12) status. Another cluster assigned to microglia included co-localised CD68 and GAD1 signals (cluster_15).

Single neuronal subpopulations of excitatory and inhibitory neurons were identified for almost each of the neuronal markers used, with only RORB (cluster_5, cluster_8, cluster_10, cluster_11) and GAD1 (cluster_9, cluster_31, cluster_32, cluster_33) populations further subclustering based on co-expression of additional markers. Pan-neuronal markers were only weakly expressed in the neuronal clusters expressing other neuronal markers. Clusters exclusively expressing pan-neuronal markers (including of the biggest clusters, cluster_1, together with cluster_7 and cluster_17) identified neuronal subpopulations that could not be further defined (hence, indicated as unclassified neurons). Only clusters with strong expression of ADARB1 and LHX6, and without glial markers, were classified as inhibitory or excitatory neurons, although some glial cell populations also weakly expressed these markers (e.g., OLIG2+ [cluster_0], Iba1+ [cluster_12]). Markers for oligodendrocytes, astrocytes and microglia showed little co-localisation with neuronal clusters, apart from S100B in cluster_14 (SST+). Cluster_4, which included cells expressing all the excitatory neuron markers (RORB, FOXP2, GPC5, PCP4, LMO3, CUX2) and some inhibitory ones (ADARB1, LHX6, VIP), was localised to L1 (Fig. 2d), suggesting that it represented non-specific signal from the meninges and was therefore excluded from all analyses.

## Evaluation of accuracy and precision of IMC antibody panel

The quality and relative intensity of antibody signals in IMC compared to the IF gold standard was used to qualitatively determine the accuracy of the IMC immunostaining. To estimate the precision of our IMC antibody panel in detecting neurons, we performed an independent IMC experiment with the full the antibody panel, acquiring 3 ROIs from three different sections of MTG from the same CtrlTREM2 brain. Images were processed with SIMPLI using parameters consistent with those used throughout and detected nuclei were clustered at 1.2 resolution. 24 clusters were identified (Figure S4.a). The clusters showed similar contributions from each of the technical replicates (Figure S4.b). Marker expression per cluster was inspected (Figure S4.c) and used for assigning clusters to cell types (Figure S4.d). The coefficient of variation (CV) of the number of cells per cluster was calculated as the standard deviation of dataset divided by the mean of dataset[87] (Figure S4e, f).

The average coefficient of variation (CV) across ROIs for the neuronal clusters was $29.70 \pm 10.90\%$ ($30.06 \pm 11.56\%$ across all clusters) (Figure S4.e). The CV between averaged ROIs from three technical replicates in serial sections was $13.73 \pm 8.32\%$ for neuronal clusters ($15.30 \pm 7.60\%$ for all of the clusters together) (Figure S4.f). Power estimates based on the CV for cluster_8 (24.4%) suggested that 8-9 samples were needed to detect differences between sample groups.

## Analysis of cell and plaque densities in cortical layers

SIMPLI output data was further analysed using RStudio (version 2023.06.2 + 561) to determine cells and plaques distribution.

Cell assignment to cortical layers was performed by analysing the Y coordinate of each identified nuclei (available from the.txt Hyperion output file) representing their localisation along the long axis (thickness) of the cortex. Reference cortical layers thicknesses[88] were proportionally applied to all ROIs based on each ROI highest Y coordinate value representing the total cortical thickness (Fig. 2d). Layer areas in μm² for each ROI were calculated by multiplying relative individual layer thickness by highest X coordinate of corresponding ROI (representing its total width), then transformed to mm². Cell density in each layer was calculated dividing cell number in each layer by the corresponding layer area.

Annotations of plaque location within each cortical layer were performed by analysing the Y coordinates of pixels forming the segmented plaque masks (Figure S8.i; for plaque masking see below). Plaque density per cortical layer was calculated by dividing total plaque mask pixel area in each layer by the corresponding layer area.

## Quantification of marker+ pixel areas

The SIMPLI-based pixel area quantification was used to quantify total marker+ area within ROIs and was applied for determining total NTNG2 and synaptophysin expression (Figure S5.a), total amyloid-β+ and pTau+ deposition (Figure S6.b-e) and PLP1 (myelin) expression (Figure S8.e)

## Quantification of cells positive for intracellular amyloid-β and pTau

Intra-cellular signal of pathological proteins (Figure S6.f,g) was determined from SIMPLI output data for each segmented neuron. Positivity of intracellular immunostaining for Aβ and pTau was determined based on the mean signal intensity of Aβ and pTau in each segmented cell using a consistent, manually defined threshold across all sections (0.03 and 0.2 for Aβ and pTau, respectively). The results of this binary classification are described as a % positive for all neurons (Figure S6.h-k) or in each neuronal cluster (Fig. 4a, c).

## Cell-cell spatial interaction analysis

A Seurat object was generated from the SIMPLI clustering output data and converted to a spatial object using the XY coordinates of each detected nuclei included in the SIMPLI output file. The 2D k-nearest neighbours of nuclei included in the neuronal and glial clusters were detected using the buildSpatialGraph function included in the imcRtools R/Bioconductor package[89] (version 1.0.2). Pairwise interactions between nuclei of each cluster are represented as the sum_sigval value generated through the testInteractions function also included in

the same package (Fig. 5d,e; Figure S8.f,g). This value represents the contrast of a nearest neighbour cell type/cell type interaction count against the null distribution generated by permuting all cell groups. Sum_sigval therefore provides an index of likelihood that different cell types are significantly more frequently found adjacent to each other (sum_sigval = 1) or not (sum_sigval = −1).

### Plaque masking

Plaque area and a 50 μm surrounding annulus were separately segmented using an ImageJ macro created in house applied on the Aβ channel of each ROI. This macro performed despeckle, filtering of particles below 200 μm and Gaussian blur (sigma = 5) of the Aβ$^+$ signal. Masking of the resulting image identified plaques (Figure S8.i; Figure S8.k,m) and enlargement of the outer border was used to define a surrounding annulus. Cell locations within the plaque or annulus then were computed based on their XY coordinates (Figure S8.j,l).

### Generation of single nucleus RNA transcriptomic dataset

The snRNAseq dataset used here has been described in a preprint[52]. It includes data from nuclei isolated from frozen MTG blocks from homologous regions in the contralateral hemisphere paired to the FFPE sections used for the IMC studies with blocks from the MTG of 10 additional brains (4 control samples, one of which carrying the *TREM2* risk variant *R62H*, and 6 AD samples, 5 of which were heterozygotic for the *TREM2* risk variant [3x *R62H*, 2x *R47H*]); Supplementary Table 4).

### Pre-processing and quality-control of snRNA sequencing data

Alignment and demultiplexing of raw sequencing data was performed using 10X Genomics Cell Ranger v3.1, with a pre-mRNA GRCh38 genome reference[90] including both introns and exons. Downstream primary analyses of gene-cell matrices were performed using our scFlow pipeline[91]. Ambient RNA profiling was performed using emptyDrops with a lower parameter of <100 counts, an alpha cut-off of ≤0.001, and with 70,000 Monte-Carlo iterations[92]. Cells were filtered for ≥400 and ≤40000 total counts and ≥200 and ≤20000 total expressive features, where expressivity was defined as a minimum of 2 counts in at least 3 cells. The maximum proportion of counts mapping to mitochondrial genes was set to 5%. Doublets were identified using the DoubletFinder algorithm, with a doublets-per-thousand-cells increment of 8 cells (recommended by 10X Genomics), a pK value of 0.005, and embeddings were generated using the first ten principal components calculated from the top 2000 most highly variable genes (HVGs)[93].

### Integration, clustering, and visualization of snRNA sequencing data

The linked inference of genomic experimental relationships (LIGER) package was used to calculate integrative factors across samples[94]. LIGER parameters used included: k: 50, lambda: 5.0, thresh: 0.0001, max_iters: 100, knn_k: 20, min_cells: 2, quantiles: 50, nstart: 10, resolution: 1, num_genes: 3000, centre: false. Two-dimensional embeddings of the LIGER integrated factors were calculated using the uniform-manifold approximation and projection (UMAP) algorithm[93] with the following parameters: pca_dims: 30, n_neighbours: 50, init: spectral, metric: euclidean, n_epochs: 200, learning_rate: 1, min_dist: 0.4, spread: 0.85, set_op_mix_ratio: 1, local_connectivity: 1, repulsion_strength: 1, negative_sample_rate: 5, fast_sgd: false. The Leiden community detection algorithm was used to detect clusters of cells from the UMAP (LIGER) embeddings; a resolution parameter of 0.0001 and a k value of 50 were used[95].

### Assigning cell type labels to snRNAseq cells

Automated cell-typing was performed essentially as previously described using the Expression Weighted Celltype Enrichment (EWCE) algorithm in scFlow against a previously generated cell-type data reference from the Allen Human Brain Atlas[91,96]. The top five marker genes for each automatically annotated cell-type were determined using Monocle 3 and validated against canonical cell-type markers[97].

### Re-clustering of the snRNAseq neuronal sub-population

Excitatory and inhibitory neuronal clusters were first sub-setted from the single cell object generated in the initial clustering and cell type annotation. Data was first normalized and scaled using Seurat's NormalizeData and ScaleData functions respectively. RunPCA function was used to calculate the first 20 PCs using the top 2000 highly variable genes. Individual samples were re-integrated with Harmony[98], using Seurat's RunHarmony() function (group.by.vars = "manifest"). To produce the final UMAP (Fig. 6a), we used the following parameters in RunUMAP() (dims = 1:20, n.epochs = 500). To identify clusters, we used first the function FindNeighbors() (dims = 1:20) and then performed unbiased clustering by using FindClusters() (resolution = 0.5). Automated cell-typing was performed essentially as previously described using the Expression Weighted Celltype Enrichment (EWCE) algorithm in scFlow against a previously generated cell-type data reference from the Allan Human Brain Atlas[91,96].

### Matching of IMC and snRNAseq neuronal clusters

For matching IMC and snRNAseq neuronal clusters, all non-neuronal IMC clusters and markers as well as genes highly expressed in all snRNAseq neuronal clusters (MAP2, APP, RBFOX3 and SYP; Figure S9a) were excluded from scoring. Average levels of expression of neuronal markers used in the IMC experiment were extracted from snRNAseq neuronal clusters (derived from read numbers). Average expression of markers for snRNAseq and IMC neuronal clusters were scaled separately to a comparable range (−2 to 5). The seven most highly expressed markers in each snRNAseq and IMC neuronal cluster were identified. The summary similarity score for each IMC-snRNAseq clusters combination (Supplementary Data file 1; Figure S9b) was calculated by multiplying the following four scores: "scaled IMC marker expression product" (product of the scaled expression within the IMC dataset of the shared top 7 markers), "scaled marker transcript expression product" (product of the scaled expression within the snRNAseq dataset of the shared top 7 markers), "shared top marker weighting" (defined as 2 or 1 when the most expressed marker in both clusters is identical or not, respectively) and "shared size quantile weighting" (defined as 2, 1.75, 1 or 0.5 when clusters are in the same size quartile or one, two or three quartiles apart, respectively). The highest similarity score for each IMC-snRNAseq clusters combination was used to identify matching clusters (Supplementary Table 5; Fig. 6b).

### Dirichlet modelling of relative snRNAseq cell-type composition

We used a Dirichlet-multinomial regression model[99] to identify changes in relative cell-type composition between AD and control groups stratified by *TREM2* variants, adjusting for additional covariates (e.g., sex, age and *APOE* and *CD33* risk genotypes). Threshold for significance was set at an adjusted *p*-value < 0.05.

### Differential gene expression analysis

We used model-based analysis of single-cell transcriptomics (MAST) to identify genes differentially expressed (associated) with quantitative measures of different histopathological features (using 4G8 amyloid, and PHF1), using each feature as a dependent variable in a zero-inflated regression analysis using a mixed-model[100]. Additionally, diagnosis (control, AD) was used as a dependent variable to identify DGE between experimental groups. Models were fit separately for each cell-type, with or without stratification by *TREM2* genotype (none, *TREM2var*, *R47H*, or *R62H*). The model specification

was zlm (~dependent_variable + (1|sample) + cngeneson + pc_mito + sex + age + APOE + CD33, method = "glmer", ebayes = F). The fixed-effect term cngeneson is the cellular detection rate as previously described, and pc_mito accounts for the relative proportion of counts mapping to mitochondrial genes. Each model was fit with and without the dependent variable and compared using a likelihood ratio test. Genes expressed in at least 10% of cells (minimum of 2 counts per cell) were evaluated for gene expression. The threshold for significant differential gene expression was a log2 fold-change of at least 0.25 and an adjusted $p$-value < 0.05.

## Trajectory analysis

Braak pseudo-time trajectory analysis was performed to infer the phenotypic transitions happening from early to late Braak stages. Unsupervised single-cell trajectory analysis was performed with *Monocle3*, an algorithm that allows to learn the sequence of gene expression changes each cell must go through as part of a dynamic biological process. The trajectory was defined on the total set of nuclei gene expression for Braak 0-2 (non-diseased) and Braak 5-6 (AD) tissues, with an origin defined by the former. We used *SeuratWrappers* to convert our *Seurat* object into a cds (CellDataSet) object with *as.cell_data_set()*. We pre-processed the cds object with num_dim=20, then performed a batch correction to integrate at the sample level with *Batchelor* using *align_cds()* function. During the batch correction, we also regressed out unwanted covariates using the model residual_model_formula_str = "~ total_features_by_counts + pc_mito + sex + age". Then, a dimension reduction was performed using *reduce_dimension()* function with default settings. Finally, we ran *cluster_cells()* and *learn_graph()* (resolution = 0.001, use_partition = F, *close_loop = F*, *learn_graph_control = list(rann.k = 100, prune_graph = TRUE, orthogonal_proj_tip = F, minimal_branch_len = 10, ncenter = 300)*) to learn the trajectory. To identify the genes differentially expressed along the trajectory we used *graph_test()* function. We then grouped the differentially expressed genes into modules by *find_gene_modules()* function with resolution = 0.05, umap.metrix=Euclidean, umap.min_dist=0.3 and umap.n_neighbours=50 L.

## Impacted pathway analysis

Impacted pathway analysis (IPA) was performed essentially as previously described using the enrichR (v 3.3) package[101,102]. Statistically significant differentially expressed genes were submitted for IPA with the over-representation analysis (ORA) enrichment method against the 'GO_Biological_Process', 'GO_Cellular_Component', 'GO_Molecular_Function', and 'KEGG' databases. Genes that are expressed in at least 5% nuclei of the neuronal clusters were used as the background gene set in the enrichment analysis. The false-discovery rate (FDR) was calculated using the Benjamini-Hochberg method, and filtering was applied at a significance threshold of ≤0.05. Pathways were then selected based on the removal of pathways with less than 3 over-lapping genes and an odds ratio below 8. Pathways specific to other cell types, over-lapping descriptions and non-specific descriptions were also removed.

## Design of figures

Immunofluorescence and IMC images were processed with ImageJ (version 2.14.0/1.54 f). Orthogonal projections and 3D reconstruction were created using "orthogonal view" function and "3D viewer" plugin, respectively, available in ImageJ. Plots were generated with RStudio (version 2023.06.2 + 561) and assembled with Adobe Illustrator (version 28.0). Boxplots in Figs. 2, 3, 5; S4, S5, S7 show median (middle line), interquartile range (box) and variability outside of first and third quartile (lines extending from box). Graphics in Fig. 1a, d were created in BioRender (Caramello, A. (2025) https://BioRender.com/z22okvn)[103].

## Statistics and reproducibility

Statistical analyses were performed in RStudio. For identifying changes in clusters proportions between sample groups (CtrlCV, CtrlTREM2, AlzCV, AlzTREM2, AlzR62H, and Alz47H, depending on the analysis), we used Dirichlet-multinomial regression[99], which accounts for differences between groups in the total numbers of cells captured (Figs. 3a, c, e; 5a, b; S8a–d). For all other analyses, we first tested the normal distribution of data using Shapiro-Wilk test. Analysis of group variance and pair-group comparisons was then performed with a two-way Wilcoxon signed-rank test when comparing two groups and ANOVA and Tukey tests or Kruskal–Wallis and two-way Wilcoxon signed-rank test when comparing four or five groups with normal and non-normal distribution, respectively. When more than two groups were tested, $p$ values of pair-group comparisons (from a Tukey test after an ANOVA test and a Wilcoxon signed-rank test after a Kruskal–Wallis test) were adjusted for multiple comparisons using the FDR method. To normalise data distribution and reduce variance, data expressed as a percentage was transformed with *arcsin* before proceeding with normality and statistical tests. $P$ values are indicated as: non-significant, ns, $p > 0.05$; *$p ≤ 0.05$; **$p ≤ 0.01$; ***$p ≤ 0.001$; ****$p ≤ 0.0001$. Sample size was determined based on both power calculations described above and the availability of variant *TREM2* brain samples. No data were excluded from the analyses. The experiments were not randomized and the investigators were not blinded to allocation during experiments and outcomes assessments. However, image analyses were performed automatically on all images with the same settings to reduce potential bias. All IMC raw images generated in this study and R code[104] used for data analysis are available on Figshare (https://doi.org/10.6084/m9.figshare.27909663.v1; https://doi.org/10.6084/m9.figshare.27901113.v1) and can be used to reproduce all analyses shown in this study.

## Reporting summary

Further information on research design is available in the Nature Portfolio Reporting Summary linked to this article.

# Data availability

Source Data for all plots are provided. Raw IMC images and their normalised and pre-processed version generated with SIMPLI, as well as images generated upon nuclei channel segmentation in SIMPLI, all generated in this study, are available to download from Figshare (https://doi.org/10.6084/m9.figshare.27909663.v1). The two original datasets generated by SIMPLI (area_measurements.csv and clustered_cells.csv), containing all the IMC data shown in this study, are also available to download from Figshare (https://doi.org/10.6084/m9.figshare.27901113.v1). The snRNAseq dataset analysed in this study (MTG samples only) is available to download (Synapse ID: syn36812517; GEO ID: GSE297004, https://www.ncbi.nlm.nih.gov/geo/query/acc.cgi?acc=GSE297004). Source data are provided with this paper.

# Code availability

Analysis of snRNAseq dataset was performed using the scFlow pipeline[91]. R scripts for analysing the SIMPLI output files generated from the automated analysis of IMC images and for matching IMC and snRNAseq clusters are available on a dedicated GitHub repository[104] (https://github.com/AlessiaCaramello/Vulnerable-neurons-in-AD).

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

## Acknowledgements

We thank the donors and their families for the use of human brain tissue in this study and the UK brain bank staff for making it available. Tissue samples were provided by the London Neurodegenerative Diseases Brain Bank at King's College London. The brain bank receives funding from the UK Medical Research Council and, as part of the Brains for Dementia Research programme, is jointly funded by Alzheimer's Research UK and the Alzheimer's Society. Tissue for this study was also provided by the Newcastle Brain Tissue Resource which is founded in part by a grant from the UK Medical Research Council (G0400074), by NIHR Newcastle Biomedical Research Centre and Unit awarded to the Newcastle upon Tyne NHS Foundation Trust and Newcastle University, and as part of the Brains for Dementia Research Programme jointly funded by Alzheimer's Research UK and Alzheimer's Society. In addition, tissue was provided by the Queen's Square Brain Bank, UCL. Tissue samples were supplied by The Manchester Brain Bank, which is part of the Brains for Dementia Research programme, jointly funded by Alzheimer's Research UK and Alzheimer's Society. We acknowledge the Oxford Brain Bank, supported by the Medical Research Council, the NIHR Oxford Biomedical Research Centre and the Brains for Dementia Research programme, jointly funded by Alzheimer's Research UK and Alzheimer's Society. Tissue samples and associated clinical and neuropathological data were supplied by Parkinson's UK Brain Bank at Imperial, funded by Parkinson's UK, a charity registered in England and Wales (258197) and in Scotland (SC037554). We also acknowledge the South West Dementia Brain Bank at the University of Bristol, which provided tissue for this study and is currently supported by BRACE and by Alzheimer's Research UK and Alzheimer's Society through Brains for Dementia Research. We are grateful to Dr Diana Benitez for her support in the human tissue ordering and management, Dr Michele Bortolomeazzi for his help with SIMPLI troubleshooting and Alan Murphy for his help in data analysis in R Studio. PMM acknowledges generous personal support from the Edmond J Safra Foundation and Lily Safra, an NIHR Senior Investigator Award and the Rosalind Franklin Institute. This work was supported by a Cross Centre research award to PMM and JH and programme funding to PMM from the UK Dementia Research Institute, which receives its funding from UK DRI Ltd., funded primarily by the UK Medical Research Council and funding from Biogen to PMM and JSJ. AC salary and consumables were supported by the UK DRI Cross-centre post-doc programme and Alzheimer's Society (Grant Reference number: 628).

## Author contributions

P.M.M. designed the study. JSJ sourced and coordinated analyses of the tissue samples. A.C. developed and validated the antibody panel and performed all IF and IMC experiments and corresponding image and data analysis. C.T. and M.E. performed the microglia immunostaining and analysis on glial cells distribution around plaques. V.C. designed the ImageJ macro for analysis of plaques distribution E.A., M.P. and N.W. generated the scRNAseq dataset. NF performed the computational analyses on the snRNAseq dataset. PMM and JH supervised the study. AC and PMM drafted and edited the manuscript with input from all authors, each of whom reviewed the full manuscript.

## Competing interests

This study was partly funded by Biogen. PMM has received consultancy fees from Sudo Biosciences, Ipsen Biopharm Ltd., Rejuveron Therapeutics, Nimbus Therapeutics and Biogen. He has received honoraria or speakers' fees from Novartis and Biogen and has received research or educational funds from BMS, Biogen, Novartis, Invicro and Nimbus Therapeutics. The remaining authors declare no competing interests.
