## [Transparent Peer Review file · Nature Communications]

Intracellular accumulation of amyloid- β is a marker of selective neuronal vulnerability in Alzheimer's disease

Corresponding Author: Professor Paul Matthews

Version 0:

Reviewer comments:

Reviewer #2

(Remarks to the Author)

The report, transferred with reviewer responses from Nature Neuroscience, addresses roles of major pathogenic factors as determinants of differential neuronal vulnerability in late onset AD brain. Single nucleus transcriptomics combined with multiplex IHC spatial profiling are used to discriminate relatively vulnerable versus resistant populations in AD cortex based on their levels of accumulation of abeta or AT8-ptau immunoreactivity. An array of advanced bioinformatic analyses is applied to identify clues to metabolic pathways associated with resilience.

The previous reviewers expressed concerns about over-interpretations of the data, limited biological validation of inferences drawn from informatic approaches, and claims from transcriptomic data that rely heavily for their validation on uncited earlier literature. The reviews do reflect appreciation of the extensive advanced bioinformatic analyses applied to human brain as the gold standard for the study of AD.

The authors respond to most of the concerns raised with clarifications and minor modifications of their data presentation, which has somewhat improved parts of the report. Added citations to some earlier relevant literature on intracellular A β accumulation and proteostasis deficits in AD provide better context and support for intracellular A β being a vulnerability factor. The authors' response walks back a major claim in the report that hyper-phosphorylation of a single ptau is a determinant, rather than being just one possible correlate of resilience to cell death. However, their retaining a claim in the Abstract that one cell subtype accumulating pTau IR but not A β is relatively resilient is misleading, given the resilience of other cells in the population lacking the AT8 tau epitope. The report provides preliminary glimpses of how new spatial/omic methodologies could be applied to human postmortem studies in the future and an appreciation of population differences in these markers and TREM2 genotype on general vulnerability.

The adequacy of the responses to some of the original major concerns falls short and key issues need to be more fully addressed before publication in a highly visible journal. Several previous reviewers commented about the low presentation quality of images that are the basis for the automated quantitation of the antibody positivity of neurons. These are so low a magnification (and pixelated when enlarged) that the nature of the subcellular localization of A β or tau or even the intracellular versus extracellular localization cannot be easily determined. In response, the authors provided "close-up" images, which are roughly at the same low magnification and only compare the gross pattern of labeling with different antibodies. As pointed out in the reviews, it would be valuable to clearly define how positivity is determined for A β , pTau, and neuronal identity. With appropriate magnification, the images could provide important validation for their conclusions about disease stage-associated pathway enrichment based on pseudo-time trajectories. The authors' explanation for a threshold of immunoreactivity that discriminates neurons with "100% A β positivity" as vulnerable relative to those with 50% or less raises several possible questions, both methodological and biological, but at minimum, the methodology and its validation need to be discussed in the report and not just in the Reviewer response, which currently appears to be the case.

For the transcriptomic analyses of clusters, the authors have relied heavily on enrichment analyses of GO/KEGG pathways, the constituents of which are not logically assigned in some cases as originally raised as an issue in the first review. A further issue is the use of the Enrichr tool for enrichment analyses which implements Fisher's Exact Test (FET) or hypergeometric test to identify if a given gene set is statistically significantly enriched in a test gene set or not. Although the general use of Enrichr for enrichment analyses is not technically wrong, the problem with this tool is it uses total protein coding genes of about 20,000 by default which yield inflated FDR (very low FDR values).

In the example from the earlier review, a conclusion (Line 621-624) drawn from the enrichment analysis in the Braak pseudo-time trajectory analysis is that autophagy is enriched early and contributes to neuronal vulnerability. In Fig. 6 f,g, the authors classify five sub-categories under a broadly defined term "autophagy" based on an unknown definition of the

autophagy category that was questioned as to its validity in the initial review. The connection of at least 4 of these to autophagy function is unclear. One of the other two sub-categories shown to be enriched in Exc-L4-6-RORB-LCN15 (Selectively vulnerable neurons) is Protein Autoubiquitination. This category consists of 72 genes and the authors found only 4 genes to be present in these neurons (RNF11, UBE2B, UBE3A, TRIM37). Despite this, the reported FDR value is very low (i.e., 0.009733) which can be attributed to the much larger background number of genes used by Enrichr. The best established subcategory of autophagy, is "Regulation of Macroautophagy" which, despite having only 3 genes in these neuronal population out of 133 category genes, Enrichr reports it to be highly statistically significant with FDR value of 0.004318. This phenomenon is consistent across the many categories reported. The ideal background would have been the total number of unique genes identified in the snRNAseq study which would be much smaller and therefore, they would not yield significance for many of these categories. One cannot use the genes as background that are not expressed in brain or those not identified in the study for enrichment analyses.

A further conflict to the authors' conclusions regarding autophagy is that the processes classified under the term "autophagy" in Fig. S8.b are also enriched in Exc-L6-FEZF2 which is matched to cluster_16 and is "unaffected" in AlzCV compared to CtrlCV (Fig. S3.b). The authors claim that another category assigned to autophagy "Regulation Of Tau-Protein Kinase Activity" is enriched (HSP90AA1, HSP90AB1, CLU = 3 genes out of 10 in the category) in Exc-L4-6-RORB-LCN15 (selectively vulnerable). However, supp. Table 9 shows that this sub-category is also enriched in Exc-L6-THEMIS-LINC00343 (selectively unaffected) neurons, with an even lower FDR.

Given the questionable relevance of subcategories assigned to "autophagy", the Enrichr -related inflation of significance, and exceptions to the claims in the profiles of unaffected neurons, there is minimal validity to the claim in the pseudo-time analysis that autophagy is down-regulated at early AD stages. Several earlier findings (eg. PMID: 27813694) actually report autophagy upregulation at early AD stages. It is also unclear why the authors single out autophagy processes while ignoring to mention (as least for the conclusion) other interesting and well-defined categories like glycolytic and synaptic processes (Exc-L4-6-RORB-LCN15), Chloride channel activity and (again) glycolytic processes in (Exc-L5-RORB-LINC01202) which are changing early in the pseudo-time analysis.

MINOR:

1. Line 603: "to define modules representing the early, mid and late stages of the (Fig. 6.e; Fig. S8.c)". There is no Fig. S8.c.
2. Labels in some of the figures especially Figures S8 (Both x and y-axis) are hard to read to be able to relate to the text. The text especially related to the snRNAseq clusters enrichments and trajectory analysis make hardly one reference to the figure/table they are related to. The authors need to relate each or most observations in the text that are related to figure/table especially when such complex information is being conveyed.
3. It appears that the authors have used two different databases within Enrichr which may give slightly different results. For e.g., "GO_Biological_Process_2023" was used for results related to supp table 8 while "GO_Biological_Process_2021" was used for results related to supp table 7. It is highly recommended to use to latest GO categories for consistency when complex multi-dimensional data is involved for more accurate cross-comparison.
4. The authors claim, "When regressed for greater β -amyloid deposition, this neuronal subtype showed decreased expression of pathways for intracellular protein transport between endoplasmic reticulum and Golgi, which are involved in the response to unfolded proteins and autophagic protein degradation" (Line 571-573) when reporting enriched pathways in Exc-L4-6-RORB-LCN15 cluster matching selectively vulnerable RORB+FOXP2+ neurons. This is incorrect since it is "Late endosome to golgi transport" which is downregulated in these neurons as reported in supp table S8 as well as supp Fig. S8 when regressed for amyloid beta. In fact, the "regulation of ER to Golgi vesicle-mediated transport" is downregulated in Exc-L6-THEMIS-LINC00343 CV, which is the "unaffected" neuronal subtype when regressed for "diagnosis".

Reviewer #3

(Remarks to the Author)

The revised version and responses submitted by Caramello et al aim to address concerns raised by Rev 1. They address previous raised issues with respect to cell count/density statistics; intracellular amyloid and Tau aggregation in selective neuronal populations and the absence of sufficient efforts for validation or mechanistical implications of the descriptive findings.

Overall, this is a well described and well performed spatial biology study that provides novel insights in neuronal populations that are vulnerable in AD, while others are resilient. Moreover, they tie cell loss/resilience of those populations together with intra cellular Tau and amyloid as well as an associated glial response. Finally, snRNAseq provides some mechanistic clues showing differential gene enrichment patterns for those neuronal populations.

Most points raised by the reviewesr 1+3 have been addressed.

Below are the remaining points that require attention.

- Fig 3: Previously, concerns were raised about the cell count/density statistics. The authors assured that any disagreements in between the plots were a plot error and submit revised versions of the plots as well as cell density stats across all cortical layers. What is the accuracy and precision of the IMC panel? How many technical replicates per patient (not per section) were collected? What is the %CV of the method within individuals?

- Some concern remains regarding the observed differences which appear as stat. significant in one analysis but lost for some densities when including more groups, while not affected for others. Eg cluster 10 (RORB/GPC5) neurons are higher in L6 in Ad as compared to Ctrl (Fig 3b) though this significance is lost in Fig 3d (L6) upon inclusion of CtrlTrem2 and

ADTrem2. Conversely, significance is maintained for difference in cluster 32 neurons (GAD1/FOXP2) (L3+L6, Fig 3b) vs (L3+L6, Fig 3d).

- The spreads observed for certain populations (with low numbers) shows a rather arbitrary pattern showing significance for some but not for others. Please motivate why different statistical tests have been used for two group comparison of either the densities (Wilcoxon, Fig 3b) or counts (Dirichlet regression, 3a).

- While the changes that appear significant might be true, this would call for some validation using immunohistochemistry along with high res IMC images. Keeping in mind that antibodies might interact with each other, examples of specificity of selecting distinct neuronal populations with these markers should be provided such using straightforward IHC.

- Fig 4: The second part of Fig4 (f-h) has been removed. Please explain why. The intensity scale in 4c does not allow to identify whether there are changes in intra.Ab for cluster 8 and cluster 32 neurons in between groups. Please adjust or present boxplots for those.

- Fig 4: The authors define intracellular AT8 pos Tau as NFT. Please provide some structural means (eg ThT or and PHF) to validate that this is fibrillar tau.

- Fig S5: The authors show that Moab2 stains intracellular Abeta. The study would benefit from some additional stainings/images using this approach to validate the IMC findings such as for eg the GAD1+ or RORB+ neurons, demonstrating changes in intracellular Abeta across different groups/Braak stages.

- Fig 5: The 3D reconstruction is difficult to discern and should be increased.

- Fig 5/Line 402: The authors interpret their findings of CD68/GAD1 double positivity that those microglia phagocytose GAD1+ neurons. Again, some high res imaging results showing how activated microglia phagocytose those neurons would enhance this part of the study.

- Line 448: The authors show that S100+ astrocytes spatially interact with RORBMAP2 and GAD1ADARB1 neurons. They conclude however that astrocytes associate spatially with resilient neurons, which would be cluster 10 (RORB1GPC5).

- line 484: the authors refer to a preprint describing a similar snRNAseq study. please detail potential overlap.

Version 1:

Reviewer comments:

Reviewer #2

(Remarks to the Author)

The authors have been very responsive to the issues raised in the earlier review. The revised MS has addressed all of my earlier concerns in a satisfactory manner. There is need for one correction on page 27.

Results, page 27:

"Chloride channel-related pathways were enriched at the early or mid stages of disease in both Exc-L5-RORB-LINC01202 and L6-THEMIS-LINC00343 clusters, (e.g., "chloride channel activity" in Modules 7 and 4 modules, respectively; Fig.6.g, Fig.S10.e). Chloride channels regulate neuronal excitability. Their homeostasis is altered in AD and their restoration has been suggested as a potential target for reversing cognitive decline previously."

The modules numbers need to be corrected. They should be Modules 3 and 2, respectively instead of 7 and 4.

(Remarks on code availability)

Reviewer #3

(Remarks to the Author)

Rev: The authors made a substantial effort to clarify the previously raised comments. Some minor aspects however should be further clarified. Please see my comments in red in line with authors responses.

- Fig 3: Previously, concerns were raised about the cell count/density statistics. The authors assured that any disagreements in between the plots were a plot error and submit revised versions of the plots as well as cell density stats across all cortical layers. What is the accuracy and precision of the IMC panel? How many technical replicates per patient (not per section) were collected? What is the %CV of the method within individuals?

The reviewer raises appropriate questions regarding the reliability of the quantitative IMC analyses.

The first question is accuracy. We determined accuracy of IMC relative to an immunofluorescence (IF) “gold standard”. Neuronal antibodies added to the panel were first validated with IF relative to published examples (provided by the manufacturer) of prior uses of the antibodies and in conjunction with established pan-neuronal or synaptic markers. The quality and relative intensity of antibodies’ signals in IMC compared to this IF “gold standard” was used to qualitatively determine the reliability of the IMC immunostaining. We have added a new supplementary figure with IF and IMC images of all new neuronal antibodies that passed these quality control steps and were used in this study (Fig.S1).

Rev: While this reviewer appreciates the effort, it is hard to discern from the Figure and the legend what is what and how the IF signal compares to the IMC. I presume you aim to show a relative/qualitative assessment of staining pattern, i.e. nr of pos. staining in rel. comparison to the panneuronal marker. This should be clarified in the methods and the legend. Also, specify in the legend that the Ir191 signal is a nuclear marker and what metals were used for the respective antibodies.

The reliability of our antibody panel as well as our method for detecting and classifying neuronal and glial populations also was confirmed by comparing proportions of glial populations and excitatory/inhibitory neurons expected from prior published reports vs. those observed (Fig.2c) and the relative cortical layer locations of neuronal populations in CtrlCV samples (Fig.2d).

Rev: For 2c: I am not sure that the rel. number of oligodendrocytes corresponds to the Lit value. Please clarify this deviation.

To allow the reviewer to better appreciate our approach, we have first, as noted in the response to Reviewer 1 above, made all raw and processed IMC images a SIMPLI output files are now available for reviewers to inspect (<https://figshare.com/s/311a85639e9de2a4e502>). R scripts used for data analysis of SIMPLI output files are available in a GitHub repository (<https://github.com/AlessiaCaramello/Vulnerable-neurons-in-AD>). All data presented in this study thus can be re-generated using our R scripts using either SIMPLI output files or raw IMC images (which will require processing with SIMPLI).

Our results are based on the means of values from 3 ROIs per patient, all were selected from the same section. We expanded the methods section to clarify how ROIs were selected and acquired as described below.

Methods, page 33: “One FFPE section per sample was stained with the full conjugated antibody panel (Supp. Table 3) following the same protocol as for immunofluorescence. After overnight incubation with metal-tagged antibodies and an iridium (Ir191/193) intercalator (Standard BioTools, CA, USA) to identify nuclei, slides were washed with water 3 x 10 minutes each and air dried for 20 minutes. IMC was performed using a Hyperion Tissue Imager coupled to a Helios mass cytometer (Fluidigm, Standard BioTools, CA, USA). The instrument was first tuned using the manufacturer’s 3-Element Full Coverage Tuning Slide before the slides were loaded into the device. Three ROIs were ablated from the same FFPE section for each sample at a laser frequency of 200Hz with 1µm resolution. Each ROI was spanning the full cortical depth from L1 to L6 (grey matter – representative ROI in Fig.1.B) adding up to a total of 6 hours of acquisition per sample (~2h per ROI, corresponding to 1.2±0.15 mm² of area ablated per ROI) keeping the total area acquired per sample equal (total 3.62±0.11 mm² ablated area per sample). Data was stored as .txt files that were used for subsequent processing (Fig.1.d).”

To estimate the precision of our IMC antibody panel in detecting neuronal populations, we performed a new IMC experiment with the antibody panel used to generate the primary data in the manuscript, acquiring 3 ROIs from 3 different sections of MTG from the same patient (CtrlTREM2 – NP0122013_1, NP0122013_2, NP0122013_3). Images were processed with SIMPLI using the same settings as in our original study and detected nuclei were clustered (resolution, 1.2). 24 clusters were identified (Revision Figure 2.a), with evenly distributed contributions from all of the slides/technical replicates (Revision Figure 2.b). Marker expression in each cluster (Revision Figure 2.c) was used for assigning clusters to cell types (Revision Figure 2.d), as described in the manuscript.

The pattern of clustering was not identical to that in reported in the manuscript. This is due to the lower number of input cells which limited Seurat clustering ability despite using a higher resolution (1.2 here, compared to 0.9 in the original study).

In this new, smaller dataset, we identified 5 clusters of excitatory neurons, 6 of inhibitory neurons, 3 of unclassified neurons, 2 of astrocytes, 2 of microglia and 3 of oligodendrocytes. All glial and neuronal populations identified showed a direct correspondence with those found in our original study UMAP, e.g., the RORB+FOXP2+ cluster_18 (similar to cluster_8 in the original study) and CD68+Iba1+GAD1+ cluster_19 (similar to cluster_15 in the original study).

As fewer clusters were identified in this analysis of a small dataset compared to that in the original study (23 vs. 36), 13 clusters identified in the original study do not show a direct correspondence to clusters in this new analysis (4x excitatory neurons, 6x inhibitory neurons, 1x astrocytes and 1x oligodendrocytes). The markers of neuronal populations missing as distinct clusters in this new analysis are present in the two “positive for all” clusters (cluster_1 and cluster_23), suggesting that data from a larger number of cells might have allowed these populations to be clustered separately.

Revision Figure 2 – Clustering and cell type assignment of cells identified from 3 ROIs from 3 slides from the same sample. (a,b) UMAP plots of clusters identified in this analysis, shown by cluster identity (a) and slide/ROI origin (b) (NP0122013_1 ROIs #2,3,4, NP0122013_2 ROIs #1,2,3, NP0122013_3 ROIs #2,3,4). (c) Heatmap of intensities of IMC marker expression in each cluster and preliminary assignments to cell types. (d) Final assignment of clusters to neuronal and glial populations based on markers expression shown in (c).

We then computed the coefficient of variation (CV) of the numbers of cells in cluster generated from 3 individual ROIs in single sections from the same individuals (Revision Figure 3a). The average CV in CtrlCV samples was 30.06±11.56% for all clusters and 29.7±10.9% for neuronal clusters only, with a trend towards higher CV in smaller clusters (cluster_31 to cluster_35).

However, the precision for our study was higher than this. We acquired multiple ROIs per sample to minimise the impact of variation from ROI to ROI in the same section and better estimate the true values as the means across the comparable ROIs.

The CV between the means across 3 ROIs from 3x technical replicates decreased to $15.3 \pm 7.6\%$ for all clusters and $13.73 \pm 8.32\%$ for neuronal clusters only (Revision Figure 3b).

Revision Figure 3 – Coefficient of variation (CV) between ROIs from the section individual and between technical replicates of means across three ROIs of three sections taken from the same individual. (a) CV of cell number per cluster between ROIs of the same individual from our original study dataset, grouped by disease (Ctrl/Alz) and TREM2 variants (CV/TREM2). (b) CV of mean cell number per cluster between technical replicates of three slides from each of three different control brains (CtrlTREM2 – NP0122013_1, NP0122013_2, NP0122013_3).

Rev: This effort is highly appreciated and the Figures should go in the SI.

- Some concern remains regarding the observed differences which appear as stat. significant in one analysis but lost for some densities when including more groups, while not affected for others. Eg cluster 10 (RORB/GPC5) neurons are higher in L6 in Ad as compared to Ctrl (Fig 3b) though this significance is lost in Fig 3d (L6) upon inclusion of CtrlTrem2 and ADTrem2. Conversely, significance is maintained for difference in cluster 32 neurons (GAD1/FOXP2) (L3+L6, Fig 3b) vs (L3+L6, Fig 3d).

We understand the reviewer uncertainty. The differences in p values for the cluster contrasts described by the reviewer appear to change because the different statistical analyses were applied. When comparing two groups (CtrlCV and AlzCV) we applied Wilcoxon signed-rank test, while when comparing four groups (CtrlCV, AlzCV, CtrlTREM2, AlzTREM2) we applied ANOVA and Tukey tests or Kruskal–Wallis and Wilcoxon signed-rank test depending on whether groups showed normal or non-normal distributions, respectively. When more than two groups were tested, p values of pair-group comparisons (from Tukey test after ANOVA test and Wilcoxon signed-rank test after Kruskal–Wallis test) were adjusted for multiple comparisons using the FDR method. For example, the p value for numbers of L6 neurons in cluster 10 (RORB/GPC5) for the CtrlCV-AlzCV comparison was 0.033, but a p value of 0.214 was found for the CtrlCV-AlzCV-CtrlTREM2-AlzTREM2 ANOVA.

Similarly, the p values for cluster 32 (GAD1/FOXP2) neurons in L3 and 6 for the CtrlCV-AlzCV comparison were 0.003 and 0.005, respectively, but were 0.018 and 0.025, respectively, for the CtrlCV-AlzCV-CtrlTREM2-AlzTREM2 comparison. We have updated the material and methods section to better clarify our statistical analyses as below.

Material and methods, page 37:

Rev: This reviewer is familiar with the basics using different stat. tests such as that multiple group comparison requires multiple testing and an adjusted p-values. I suggest, the authors should rather clarify that the trends are maintained despite loss of stat. significance due to multiple testing.

- Fig S5: The authors show that Moab2 stains intracellular Abeta. The study would benefit from some additional stainings/images using this approach to validate the IMC findings such as for e.g. the GAD1+ or RORB+ neurons, demonstrating changes in intracellular Abeta across different groups/Braak stages.

We have performed further immunofluorescences staining using MOAB2/GAD1 or MOAB2/RORB in each sample group (CtrlCV, CtrlTREM2, AlzCV, AlzTREM2) to show accumulation of intraA β in both neuronal populations across Braak stages. The images were added as Fig.S6.

Rev: These images are very much appreciated. Fig S6 does not convey any relation of intracellular Abeta to Braak stages. Please specify in the text and the legend what Braak stage the images in S6 refer to. Do you have similar images for all groups? Otherwise state that these are just representative as proof of principle.

Minor comments:

- The SI Table pdf only contains Suppl Tab 5
- Legend Fig S2. Should say “ROI”

(Remarks on code availability)

REVIEWERS COMMENTS and POINT-BY-POINT RESPONSE

Reviewer #2 (Remarks to the Author):

The report, transferred with reviewer responses from Nature Neuroscience, addresses roles of major pathogenic factors as determinants of differential neuronal vulnerability in late onset AD brain. Single nucleus transcriptomics combined with multiplex IHC spatial profiling are used to discriminate relatively vulnerable versus resistant populations in AD cortex based on their levels of accumulation of abeta or AT8-ptau immunoreactivity. An array of advanced bioinformatic analyses is applied to identify clues to metabolic pathways associated with resilience.

The previous reviewers expressed concerns about over-interpretations of the data, limited biological validation of inferences drawn from informatic approaches, and claims from transcriptomic data that rely heavily for their validation on uncited earlier literature. The reviews do reflect appreciation of the extensive advanced bioinformatic analyses applied to human brain as the gold standard for the study of AD.

The authors respond to most of the concerns raised with clarifications and minor modifications of their data presentation, which has somewhat improved parts of the report. Added citations to some earlier relevant literature on intracellular A β accumulation and proteostasis deficits in AD provide better context and support for intracellular A β being a vulnerability factor.

We thank the reviewer for the appreciation of our study and its improvements following the previous round of peer-review revision. We are grateful to the reviewer for the close reading of the manuscript and thoughtful criticism.

The authors' response walks back a major claim in the report that hyper-phosphorylation of a single ptau is a determinant, rather than being just one possible correlate of resilience to cell death. However, their retaining a claim in the Abstract that one cell subtype accumulating pTau IR but not A β is relatively resilient is misleading, given the resilience of other cells in the population lacking the AT8 tau epitope.

We appreciate that our use of the term is misleading and apologise. Our intention was to highlight that neurons accumulating pTau without intraA β appeared to be "relatively resilient" to loss when compared with those neurons accumulating intraA β . However, of course, we agree with the reviewer that neurons that do not accumulate pTau⁺ or intraA β also appear resistant to loss relative to those that accumulate intraA β and see that by commenting so singularly on this population we are misleading readers.

In the revised manuscript, we have changed the text throughout to refer to the RORB⁺GPC5⁺ subtype as accumulating pTau – a description of the direct observation - rather than making the stating that they are "relatively resilient".

See, e.g., Abstract, page 2:

"By contrast, a distinct L3 RORB⁺GPC5⁺ subtype that progressively showed increase in pTau⁺ signal was not reduced in cell number."

We also now reserve use the term “resilient” for neuronal populations that do not change in number or accumulate pathological AD proteins (see references to the Exc-L5-6-FEZF2 and L6-THEMIS-LINC00343 population [l. 610]), except when referring explicitly earlier descriptions in the literature (l. 768).

There also may have been a point of misunderstanding. We did not claim in the manuscript that ptau was a “determinant” of resilience. Instead, we cited previous literature in the introduction: “Other work has suggested that accumulation of pTau may not impair neuronal function, at least initially (Kuchibhotla, K. V. et al. 2014) and may even enhance resilience to apoptosis (Li, H.-L. et al., 2007; Wu, M. et al., 2023).” We believe this is relevant to include as background but will remove the sentence if the reviewer believes that it contributes to a misleading view of our results.

The report provides preliminary glimpses of how new spatial/omic methodologies could be applied to human postmortem studies in the future and an appreciation of population differences in these markers and TREM2 genotype on general vulnerability.

We are grateful for the positive comment and believe that our observations will be of general interest.

The adequacy of the responses to some of the original major concerns falls short and key issues need to be more fully addressed before publication in a highly visible journal. Several previous reviewers commented about the low presentation quality of images that are the basis for the automated quantitation of the antibody positivity of neurons. These are so low a magnification (and pixelated when enlarged) that the nature of the subcellular localization of A β or tau or even the intracellular versus extracellular localization cannot be easily determined. In response, the authors provided “close-up” images, which are roughly at the same low magnification and only compare the gross pattern of labelling with different antibodies.

As pointed out in the reviews, it would be valuable to clearly define how positivity is determined for A β , pTau, and neuronal identity.

*As mentioned in the methods, the images generated with IMC have a resolution of 1 μ m. Although noted in the methods, this will not have been apparent in the online manuscript uploaded for review; the images provided there are of lower resolution due to file compression need for uploading to the review website. To allow the reviewer access to the full versions that will be available to readers to download as a resource, we have uploaded raw, normalised and pre-processed images as well as the segmented nuclei images on the figshare online repository, which the reviewers now can access with a private link (<https://figshare.com/s/311a85639e9de2a4e502>). Because this is a large file that takes time to download and can be complex to navigate, we also provide a single example set of raw, normalised and pre-processed images and the segmented nuclei image for one ROI from a CtrlTREM2 sample (DNA and A β 4G8 channels shown as example in **Revision Figure 1.a**). This is accessible via the private link <https://figshare.com/s/5c5a5314216fdbcbe4c1>.*

*The image analysis methods that we used are set out in the manuscript. Briefly, we identified intraA β after cell segmentation with SIMPLI (Single-cell Identification from MultiPLexed Images), software developed for cell segmentation in high-dimensional images (Bortolomeazzi et al. 2022). The neuronal A β signal overlapping with segmented nuclei makes their co-localisation clear (**Revision Figure 1.a,c** – original composite image “4G8 DNAmask.tif” from CtrlTREM2 sample available on <https://figshare.com/s/5c5a5314216fdcbbbe4c1>). Similarly, intracellular pTau, which also is co-localised with segmented neuronal nuclei, can be distinguished from extracellular pTau (see below, **Revision Figure 1.b,d** showing a composite image “pTau DNAmask.tif” from an AlzTREM2 sample available on <https://figshare.com/s/5c5a5314216fdcbbbe4c1>). We have added panels from Revision Figure 1.c,d to Fig.S5 of the uploaded manuscript.*

To provide further confidence in the neuronal intraA β co-localisations, we have added confocal imaging orthogonal projections and 3D reconstructions of the A β -42 specific antibody MOAB-2 stained together with RORB or GAD1 in Fig.S6. These new images show intracellular A β -42 signal co-localised with either RORB or GAD1, confirming that intraA β can accumulate in these vulnerable neuronal subtypes.

Revision Figure 1 – Examples of high-resolution images (provided on figshare [<https://figshare.com/s/5c5a5314216fdbcbe4c1>] with examples of intracellular A β and pTau immunostaining (double-positive cells indicated by the yellow and red arrowheads, respectively). (a) Example of normalised and pre-processed images for the DNA and A β 4G8 channels, and nuclear segmentation of the DNA channel from a single ROI of a CtrITREM2 sample generated by SIMPLI. (b) Example of pre-processed images of pTau channel and nuclei segmentation of the DNA channel from a single ROI of a AlzTREM2 sample generated by SIMPLI. (c,d) Crops from dotted yellow squares in (a) and (b) showing overlap between A β 4G8 (c) or pTau (d) channels and the DNA segmented nuclei mask, respectively. Scale bars represent 100 μ m in (a,b) and 50 μ m in (c,d).

The authors' explanation for a threshold of immunoreactivity that discriminates neurons with "100% A β positivity" as vulnerable relative to those with 50% or less raises several possible questions, both methodological and biological, but at minimum, the methodology and its validation need to be discussed in the report and not just in the Reviewer response, which currently appears to be the case.

We apologise to the reviewer for being unclear. Responding to this may benefit from a brief review of how image analysis is performed with SIMPLI. We have not elaborated on this in the manuscript methods as the details are described more fully in the original method description cited (Bortolomeazzi et al. 2022) and in the GitHub repository (<https://github.com/ciccalab/SIMPLI>).

Neuronal identity in the IMC imaging is determined from markers expressed in each neuronal subpopulation. Briefly, all IMC raw images are normalised and pre-processed with the same settings within SIMPLI pipeline to remove background (available on figshare <https://figshare.com/s/311a85639e9de2a4e502>). The DNA channel is used by SIMPLI to segment single nuclei (segmented nuclei images available on figshare <https://figshare.com/s/311a85639e9de2a4e502>). For these data, we have included the mean signal intensity for each marker in each cell used for this analysis is available in the corresponding column ("Intensity_MeanIntensity_Ab" and "Intensity_MeanIntensity_pTau") of the clustered_cells.csv SIMPLI output file (on figshare <https://figshare.com/s/cb9e0c5c3eb04ef542b5>). Markers expressed in each segmented cell are used for unsupervised clustering with Seurat within the SIMPLI pipeline (see clustered_cells.csv SIMPLI output file on figshare <https://figshare.com/s/cb9e0c5c3eb04ef542b5>; Fig.2a and Fig.S3a).

Each cell within a cluster is then assessed for co-localised A β or pTau immunostaining based on a manually validated threshold that optimally discriminated two classes of immunostaining (positive or negative; in this specific case, the threshold used consistently across all sections was set at 0.03 intensity units for A β and 0.20 intensity units for pTau).

Having thus scored each neuron in clusters as positive or negative for each of these markers, we could determine the percentage of the neurons positive for pTau (pTau mean signal intensity > 0.20 intensity units) or positive for intraA β

($A\beta$ mean signal intensity > 0.03 intensity units). This percentage is shown in Fig. 4a,c.

To make this clearer in the revised manuscript, we have extended our explanation of how intra $A\beta$ and pTau cell positivity is determined in the methods section (see below). Moreover, we have made all R scripts used in our study available in a GitHub repository (<https://github.com/AlessiaCaramello/Vulnerable-neurons-in-AD>) which can be used to rapidly re-analyse the raw data generated by SIMPLI for readers who wish to do so (clustered_cells.csv file available on figshare <https://figshare.com/s/cb9e0c5c3eb04ef542b5>). The latter will further increase the transparency of our analysis.

Methods, page 36:

“Quantification of intracellular amyloid- β and pTau signal

Quantification of cells positive for intracellular amyloid- β and pTau
Intra-cellular signal of pathological proteins (Fig.S5.f,g) was determined from SIMPLI output data for each segmented neuron. Positivity of intra-cellular immunostaining for $A\beta$ and pTau was determined based on the mean signal intensity of $A\beta$ and pTau in each segmented cell using a consistent, manually defined threshold across all sections (0.03 and 0.2 for $A\beta$ and pTau, respectively). The results of this binary classification are described as a % positive for all neurons (Fig.S5.h-k) or in each neuronal cluster (Fig.4.a,c).”

For the transcriptomic analyses of clusters, the authors have relied heavily on enrichment analyses of GO/KEGG pathways, the constituents of which are not logically assigned in some cases as originally raised as an issue in the first review.

Pathway assignment to categories was performed manually to allow better visualisation of biological processes affected. We agree with the reviewer that “autophagy” was not a well-chosen summary descriptor of the range of pathways, which include those related to ubiquitination as well as heat shock proteins and chaperones (e.g. in the “Response To Unfolded Protein”, “MHC Class II Protein Complex Binding” and “Tau Protein Binding” pathways). We agree more generally that using this type of categorisation in carries additional interpretative bias and have therefore removed “Pathway Categories” entirely from our figures (Fig.6f,g, Fig.S9, Fig.S10d,e).

A further issue is the use of the Enrichr tool for enrichment analyses which implements Fisher’s Exact Test (FET) or hypergeometric test to identify if a given gene set is statistically significantly enriched in a test gene set or not. Although the general use of Enrichr for enrichment analyses is not technically wrong, the problem with this tool is it uses total protein coding genes of about 20,000 by default which yield inflated FDR (very low FDR values).

In the example from the earlier review, a conclusion (Line 621-624) drawn from the enrichment analysis in the Braak pseudo-time trajectory analysis is that autophagy is enriched early and contributes to neuronal vulnerability. In Fig. 6 f,g, the authors classify five sub-categories under a broadly defined term “autophagy” based on an

unknown definition of the autophagy category that was questioned as to its validity in the initial review. The connection of at least 4 of these to autophagy function is unclear. One of the other two sub-categories shown to be enriched in Exc-L4-6-RORB-LCN15 (Selectively vulnerable neurons) is Protein Autoubiquitination. This category consists of 72 genes and the authors found only 4 genes to be present in these neurons (RNF11, UBE2B, UBE3A, TRIM37). Despite this, the reported FDR value is very low (i.e., 0.009733) which can be attributed to the much larger background number of genes used by Enrichr. The best established subcategory of autophagy, is “Regulation of Macroautophagy” which, despite having only 3 genes in these neuronal population out of 133 category genes, Enrichr reports it to be highly statistically significant with FDR value of 0.004318. This phenomenon is consistent across the many categories reported. The ideal background would have been the total number of unique genes identified in the snRNAseq study which would be much smaller and therefore, they would not yield significance for many of these categories. One cannot use the genes as background that are not expressed in brain or those not identified in the study for enrichment analyses.

We thank the reviewer for highlighting this. As the reviewer points out, pathway enrichment analyses in the submitted manuscript were performed using the default Enrichr background of 20,000 genes. While this has generally been accepted in other publications for this kind of exploratory analysis (see, e.g., <https://pmc.ncbi.nlm.nih.gov/articles/PMC11528003/#Sec11>, <https://www.nature.com/articles/s41467-025-56124-1#Abs1>, <https://www.sciencedirect.com/science/article/pii/S0006899324004967?via%3Dihub#s0010>), we agree that statistical significance is inflated. We therefore have redone the EnrichR analyses for the revised manuscript based on the number of uniquely expressed genes common at least 5% of nuclei (13,095 genes) in our total population of 78,676 nuclei. Additionally, we limited our analyses to clusters including a minimum of 4.6% of the total nuclei. We have updated our results (page 20-26) and figures accordingly (Fig.6f,g, Fig.S9, Fig.S10d,e). Despite this more conservative analysis, the primary results of our trajectory analysis results have not changed; we were still able to identify differentially expressed pathways related to protein degradation and glycosylation.

A further conflict to the authors' conclusions regarding autophagy is that the processes classified under the term “autophagy” in Fig. S8.b are also enriched in Exc-L6-FEZF2 which is matched to cluster_16 and is “unaffected” in AlzCV compared to CtrlCV (Fig. S3.b). The authors claim that another category assigned to autophagy “Regulation Of Tau-Protein Kinase Activity” is enriched (HSP90AA1, HSP90AB1, CLU = 3 genes out of 10 in the category) in Exc-L4-6-RORB-LCN15 (selectively vulnerable). However, supp. Table 9 shows that this sub-category is also enriched in Exc-L6-THEMIS-LINC00343 (selectively unaffected) neurons, with an even lower FDR.

Given the questionable relevance of subcategories assigned to “autophagy”, the Enrichr –related inflation of significance, and exceptions to the claims in the profiles of unaffected neurons, there is minimal validity to the claim in the pseudo-time analysis that autophagy is down-regulated at early AD stages. Several earlier

findings (e.g. PMID: 27813694) actually report autophagy upregulation at early AD stages. It is also unclear why the authors single out autophagy processes while ignoring to mention (as least for the conclusion) other interesting and well-defined categories like glycolytic and synaptic processes (Exc-L4-6-RORB-LCN15), Chloride channel activity and (again) glycolytic processes in (Exc-L5-RORB-LINC01202) which are changing early in the pseudo-time analysis.

We are grateful to the reviewer for pointing out what was simply an error and for highlighting pathways we had not described in the text but also note that there may be confusion regarding what we intended for readers to conclude from the trajectory analysis.

First, we agree that it is not helpful to discuss specific pathways (e.g., autophagy-related) without acknowledging the wider cell biology that also is highlighted. This was an error. We also have revised the sections regarding transcriptomic analyses (highlighted in the revised manuscript) to more broadly discuss examples of differentially regulated pathways and of enrichments. We have included references to the synaptic, glycolytic and chloride channel related enrichments and differential expression as helpfully suggested by the reviewer. We had not initially included reference to glycolytic pathway enrichments in the trajectory analysis report because they were similar across neuronal clusters analysed but have added this as the reviewer suggests this as an example of a generally pathologically relevant pathway enriched across clusters. We also agree with the reviewer that pathways related to chloride channel activity are exclusively enriched at early Braak pseudo-time stages in neuronal populations not affected by cell loss (Exc-L5-RORB-LINC01202 and L6-THEMIS-LINC00343 clusters).

We have modified the text as follows.

Results, page 27:

“Chloride channel-related pathways were enriched at the early or mid stages of disease in both Exc-L5-RORB-LINC01202 and L6-THEMIS-LINC00343 clusters, (e.g., “chloride channel activity” in Modules 7 and 4 modules, respectively; Fig.6.g, Fig.S10.e). Chloride channels regulate neuronal excitability. Their homeostasis is altered in AD and their restoration has been suggested as a potential target for reversing cognitive decline previously.”

Discussion, page 28:

“Our results add to the growing evidence for inflammation-induced impairments in autophagy and protein degradation as mechanisms in the initiation of cortical neurodegeneration in AD. They also highlight the importance of further understanding roles for glycosylation processes and chloride channel in early neurodegeneration.”

However, there may have been some confusion about our interpretation of enrichments in modules along the pseudo-time trajectory. These should not be interpreted as evidence for up- or down-regulation pathways – the latter interpretations are only appropriate for the differential expression analyses (against regressors associated with disease progression or categorical). The

pseudo-time trajectory enrichments simply describe pathways significantly enriched in each of the transcriptional expression modules considered independently. Exc-L4-6-RORB-LCN1 is the only cluster showing enrichment for the “Regulation Of Macroautophagy” and “Protein Autoubiquitination” pathways. Exc-L6-THEMIS-LINC00343 also shows two pathways in the “autophagy” class are enriched in an early disease stage. In reporting these results, we do not claim that these autophagy pathways are either up- or down-regulated, but only that they are enriched amongst transcripts associated with early Braak stages and therefore might play a role in determining vulnerability of the neuronal subtype. By contrast, in the independent, categorical differential expression analysis, we found and report a downregulation of the “regulation of autophagy of mitochondrion” pathway for the Exc-L4-6-RORB-LCN15 cluster (Fig. S9a).

MINOR:

1. Line 603: “to define modules representing the early, mid and late stages of the (Fig. 6.e; Fig. S8.c).”. There is no Fig. S8.c.

The figure number has been corrected.

2. Labels in some of the figures especially Figures S8 (Both x and y-axis) are hard to read to be able to relate to the text. The text especially related to the snRNAseq clusters enrichments and trajectory analysis make hardly one reference to the figure/table they are related to. The authors need to relate each or most observations in the text that are related to figure/table especially when such complex information is being conveyed.

We have increased font sizes in Figs.6, S8, S9, S10. We have added references to the specific pathway names and related figure or supplementary figures numbers. See an example below:

Results, page 26:

“Enrichments in pathways related to autophagy and protein degradation (e.g., “protein auto-ubiquitination” in Module 2, “regulation of macroautophagy” in Module 6; Fig.6.f) were associated with early disease stage in cluster Exc-L4-6-RORB-LCN15, corresponding to RORB⁺FOXP2⁺ vulnerable neurons accumulating intraA β . The “response to unfolded protein” pathway was enriched at an early stage in Module 4 from the cluster, corresponding to the RORB⁺GPC5⁺ neurons (Fig.6.g).”

3. It appears that the authors have used two different databases within Enrichr which may give slightly different results. For e.g., “GO_Biological_Process_2023” was used for results related to supp table 8 while “GO_Biological_Process_2021” was used for results related to supp table 7. It is highly recommended to use to latest GO categories for consistency when complex multi-dimensional data is involved for more accurate cross-comparison.

We thank the reviewer for spotting this inconsistency. For the revised manuscript, we have re-run the Enrichr pathway analysis for Supp. Table 8 using the more recent “GO_Biological_Process_2023” database to align the outputs. Fig.S9.b (previously Fig.S8.b) and the corresponding results section “AD-relevant pathways are differentially expressed in vulnerable neuronal subpopulations with disease progression” (Results, page 23) have been updated to reflect this.

4. The authors claim, "When regressed for greater β -amyloid deposition, this neuronal subtype showed decreased expression of pathways for intracellular protein transport between endoplasmic reticulum and Golgi, which are involved in the response to unfolded proteins and autophagic protein degradation" (Line 571-573) when reporting enriched pathways in Exc-L4-6-RORB-LCN15 cluster matching selectively vulnerable RORB+FOXP2+ neurons. This is incorrect since it is "Late endosome to Golgi transport" which is downregulated in these neurons as reported in supp table S8 as well as supp Fig. S8 when regressed for amyloid beta. In fact, the "regulation of ER to Golgi vesicle-mediated transport" is downregulated in Exc-L6-THEMIS-LINC00343 CV, which is the "unaffected" neuronal subtype when regressed for "diagnosis".

We thank the reviewer for spotting this mistake. After refining our pathway enrichment analysis using the “GO_Biological_Process_2023” database (suggested in the comment above), this pathway is no longer present and was therefore removed from the results.

Reviewer #3 (Remarks to the Author):

The revised version and responses submitted by Caramello et al aim to address concerns raised by Rev 1. They address previous raised issues with respect to cell count/density statistics; intracellular amyloid and Tau aggregation in selective neuronal populations and the absence of sufficient efforts for validation or mechanistical implications of the descriptive findings.

Overall, this is a well described and well performed spatial biology study that provides novel insights in neuronal populations that are vulnerable in AD, while others are resilient. Moreover, they tie cell loss/resilience of those populations together with intra cellular Tau and amyloid as well as an associated glial response. Finally, snRNAseq provides some mechanistic clues showing differential gene enrichment patterns for those neuronal populations.

Most points raised by the reviewers 1+3 have been addressed. Below are the remaining points that require attention.

We thank the reviewer for appreciating our study and the further improvements achieved through the previous round of revisions.

- Fig 3: Previously, concerns were raised about the cell count/density statistics. The authors assured that any disagreements in between the plots were a plot error and submit revised versions of the plots as well as cell density stats across all cortical layers. What is the accuracy and precision of the IMC panel? How many technical replicates per patient (not per section) were collected? What is the %CV of the method within individuals?

The reviewer raises appropriate questions regarding the reliability of the quantitative IMC analyses.

The first question is accuracy. We determined accuracy of IMC relative to an immunofluorescence (IF) "gold standard". Neuronal antibodies added to the panel were first validated with IF relative to published examples (provided by the manufacturer) of prior uses of the antibodies and in conjunction with established pan-neuronal or synaptic markers. The quality and relative intensity of antibodies' signals in IMC compared to this IF "gold standard" was used to qualitatively determine the reliability of the IMC immunostaining. We have added a new supplementary figure with IF and IMC images of all new neuronal antibodies that passed these quality control steps and were used in this study (Fig.S1). The reliability of our antibody panel as well as our method for detecting and classifying neuronal and glial populations also was confirmed by comparing proportions of glial populations and excitatory/inhibitory neurons expected from prior published reports vs. those observed (Fig.2c) and the relative cortical layer locations of neuronal populations in CtrlCV samples (Fig.2d).

To allow the reviewer to better appreciate our approach, we have first, as noted in the response to Reviewer 1 above, made all raw and processed IMC images a SIMPLI output files are now available for reviewers to inspect

(<https://figshare.com/s/311a85639e9de2a4e502>). R scripts used for data analysis of SIMPLI output files are available in a GitHub repository (<https://github.com/AlessiaCaramello/Vulnerable-neurons-in-AD>). All data presented in this study thus can be re-generated using our R scripts using either SIMPLI output files or raw IMC images (which will require processing with SIMPLI). Our results are based on the means of values from 3 ROIs per patient, all were selected from the same section. We expanded the methods section to clarify how ROIs were selected and acquired as described below.

Methods, page 33:

“One FFPE section per sample was stained with the full conjugated antibody panel (Supp. Table 3) following the same protocol as for immunofluorescence. After overnight incubation with metal-tagged antibodies and an iridium (Ir191/193) intercalator (Standard BioTools, CA, USA) to identify nuclei, slides were washed with water 3 x 10 minutes each and air dried for 20 minutes. IMC was performed using a Hyperion Tissue Imager coupled to a Helios mass cytometer (Fluidigm, Standard BioTools, CA, USA). The instrument was first tuned using the manufacturer’s 3-Element Full Coverage Tuning Slide before the slides were loaded into the device. Three ROIs were ablated from the same FFPE section for each sample at a laser frequency of 200Hz with 1µm resolution. Each ROI was spanning the full cortical depth from L1 to L6 (grey matter – representative ROI in Fig.1.B) adding up to a total of 6 hours of acquisition per sample (~2h per ROI, corresponding to 1.2±0.15 mm² of area ablated per ROI) keeping the total area acquired per sample equal (total 3.62±0.11 mm² ablated area per sample). Data was stored as .txt files that were used for subsequent processing (Fig.1.d).”

*To estimate the precision of our IMC antibody panel in detecting neuronal populations, we performed a new IMC experiment with the antibody panel used to generate the primary data in the manuscript, acquiring 3 ROIs from 3 different sections of MTG from the same patient (CtrlTREM2 – NP0122013_1, NP0122013_2, NP0122013_3). Images were processed with SIMPLI using the same settings as in our original study and detected nuclei were clustered (resolution, 1.2). 24 clusters were identified (**Revision Figure 2.a**), with evenly distributed contributions from all of the slides/technical replicates (**Revision Figure 2.b**). Marker expression in each cluster (**Revision Figure 2.c**) was used for assigning clusters to cell types (**Revision Figure 2.d**), as described in the manuscript.*

The pattern of clustering was not identical to that in reported in the manuscript. This is due to the lower number of input cells which limited Seurat clustering ability despite using a higher resolution (1.2 here, compared to 0.9 in the original study).

In this new, smaller dataset, we identified 5 clusters of excitatory neurons, 6 of inhibitory neurons, 3 of unclassified neurons, 2 of astrocytes, 2 of microglia and 3 of oligodendrocytes. All glial and neuronal populations identified showed a direct correspondence with those found in our original study UMAP, e.g., the RORB⁺FOXP2⁺ cluster_18 (similar to cluster_8 in the original study) and CD68⁺Iba1⁺GAD1⁺ cluster_19 (similar to cluster_15 in the original study).

As fewer clusters were identified in this analysis of a small dataset compared to that in the original study (23 vs. 36), 13 clusters identified in the original study do not show a direct correspondence to clusters in this new analysis (4x excitatory neurons, 6x inhibitory neurons, 1x astrocytes and 1x oligodendrocytes). The markers of neuronal populations missing as distinct clusters in this new analysis are present in the two “positive for all” clusters (cluster_1 and cluster_23), suggesting that data from a larger number of cells might have allowed these populations to be clustered separately.

Revision Figure 2 – Clustering and cell type assignment of cells identified from 3 ROIs from 3 slides from the same sample. (a,b) UMAP plots of clusters identified in this analysis, shown by cluster identity (a) and slide/ROI origin (b) (NP0122013_1 ROIs #2,3,4, NP0122013_2 ROIs #1,2,3, NP0122013_3 ROIs #2,3,4). (c) Heatmap of intensities of IMC marker expression in each cluster and preliminary assignments to cell types. (d) Final assignment of clusters to neuronal and glial populations based on markers expression shown in (c).

We then computed the coefficient of variation (CV) of the numbers of cells in cluster generated from 3 individual ROIs in single sections from the same individuals (**Revision Figure 3a**). The average CV in CtrlCV samples was $30.06 \pm 11.56\%$ for all clusters and $29.7 \pm 10.9\%$ for neuronal clusters only, with a trend towards higher CV in smaller clusters (cluster_31 to cluster_35).

However, the precision for our study was higher than this. We acquired multiple ROIs per sample to minimise the impact of variation from ROI to ROI in the same section and better estimate the true values as the means across the comparable ROIs. The CV between the means across 3 ROIs from 3x technical replicates decreased to $15.3 \pm 7.6\%$ for all clusters and $13.73 \pm 8.32\%$ for neuronal clusters only (**Revision Figure 3b**).

Revision Figure 3 – Coefficient of variation (CV) between ROIs from the section individual and between technical replicates of means across three ROIs of three sections taken from the same individual. (a) CV of cell number per cluster between ROIs of the same individual from our original study dataset, grouped by disease (Ctrl/Alz) and TREM2 variants (CV/TREM2). (b) CV of mean cell number per cluster between technical replicates of three slides from each of three different control brains (CtrlTREM2 – NP0122013_1, NP0122013_2, NP0122013_3).

- Some concern remains regarding the observed differences which appear as stat. significant in one analysis but lost for some densities when including more groups, while not affected for others. Eg cluster 10 (RORB/GPC5) neurons are higher in L6 in Ad as compared to Ctrl (Fig 3b) though this significance is lost in Fig 3d (L6) upon

inclusion of CtrlTrem2 and ADTrem2. Conversely, significance is maintained for difference in cluster 32 neurons (GAD1/FOXP2) (L3+L6, Fig 3b) vs (L3+L6, Fig 3d).

We understand the reviewer uncertainty. The differences in p values for the cluster contrasts described by the reviewer appear to change because the different statistical analyses were applied. When comparing two groups (CtrlCV and AlzCV) we applied Wilcoxon signed-rank test, while when comparing four groups (CtrlCV, AlzCV, CtrlTREM2, AlzTREM2) we applied ANOVA and Tukey tests or Kruskal–Wallis and Wilcoxon signed-rank test depending on whether groups showed normal or non-normal distributions, respectively. When more than two groups were tested, p values of pair-group comparisons (from Tukey test after ANOVA test and Wilcoxon signed-rank test after Kruskal–Wallis test) were adjusted for multiple comparisons using the FDR method. For example, the p value for numbers of L6 neurons in cluster 10 (RORB/GPC5) for the CtrlCV-AlzCV comparison was 0.033, but a p value of 0.214 was found for the CtrlCV-AlzCV-CtrlTREM2-AlzTREM2 ANOVA. Similarly, the p values for cluster 32 (GAD1/FOXP2) neurons in L3 and 6 for the CtrlCV-AlzCV comparison were 0.003 and 0.005, respectively, but were 0.018 and 0.025, respectively, for the CtrlCV-AlzCV-CtrlTREM2-AlzTREM2 comparison. We have updated the material and methods section to better clarify our statistical analyses as below.

Material and methods, page 37:

“Statistical analyses

Statistical analyses were performed in RStudio. For identifying changes in clusters proportions between sample groups (CtrlCV, CtrlTREM2, AlzCV, AlzTREM2, AlzR62H and Alz47H depending on the analysis), we used a Dirichlet-multinomial regression which accounts for differences between groups in total number of cells captured (Fig.3a,c,e; Fig.5a,b; Fig.S7a-d). For all other analyses, we first tested normal distribution of data using Shapiro-Wilk test. Analysis of groups variance and pair-group comparisons was then performed with Wilcoxon signed-rank test when comparing two groups only and ANOVA and Tukey tests or Kruskal–Wallis and Wilcoxon signed-rank test when comparing four groups with normal and non-normal distribution, respectively. When more than two groups were tested, p values of pair-group comparisons (from Tukey test after ANOVA test and Wilcoxon signed-rank test after Kruskal–Wallis test) were adjusted for multiple comparisons using the FDR method. To normalise data distribution and reduce variance, data expressed as percentage was transformed with *arcsin* before proceeding with normality and statistical tests. P values are indicated as: non-significant, ns, $p>0.05$; * $p\leq 0.05$; ** $p\leq 0.01$; *** $p\leq 0.001$; **** $p\leq 0.0001$.”

- The spreads observed for certain populations (with low numbers) shows a rather arbitrary pattern showing significance for some but not for others. Please explain motivation regarding why different statistical tests have been used for two group comparison of either the densities (Wilcoxon, Fig 3b) or counts (Dirichlet regression, 3a).

As noted above, Dirichlet regression was used for identifying differences in cell number per cluster among groups of samples (Fig. 3a,c,e). It is performed as a pair-wise comparison regardless of the number of groups in the analysis, and then significance is expressed as FDR after correction for the multiple comparisons. Dirichlet regression was chosen for this analysis as it accounts for the mutual dependence of changes between proportions of groups of cell types.

Conversely, the cortical distribution of clustered cells is independent of the total number of captured cells. Therefore, when only two groups are compared, we applied Wilcoxon signed-rank test (Fig. 3b), while when more than 2 groups are compared, we applied ANOVA and Tukey tests or Kruskal–Wallis and Wilcoxon signed-rank test, depending on whether groups showed normal or non-normal distributions, respectively (Fig. 3d,f).

- While the changes that appear significant might be true, this would call for some validation using immunohistochemistry along with high res IMC images. Keeping in mind that antibodies might interact with each other, examples of specificity of selecting distinct neuronal populations with these markers should be provided such using straightforward IHC.

As described above, neuronal antibodies added to the panel were first validated with immunofluorescence (IF) relative to published examples (provided by the manufacturer) of prior uses of the antibodies and in conjunction with an established pan-neuronal or synaptic marker. The quality and intensity of antibodies' signal in IMC compared to this IF "ground truth" was used to qualitatively determine the reliability of the IMC immunostaining. We have added a new supplementary figure with IF and IMC images of all new neuronal antibodies that passed these quality control steps and were used in this study (Fig.S1). The reliability of our antibody panel, as well as our method for detecting and classifying neuronal and glial populations, also was confirmed by comparing proportions of glial populations and excitatory/inhibitory neurons expected from prior published reports vs. those observed (Fig.2c) and the relative cortical layer locations of neuronal populations in CtrlCV samples (Fig.2d).

While it is true that chelated metals used for tagging the antibodies use in IMC can diffuse between antibodies once in the same mix if sufficient time is allowed. To minimise this risk, each antibody mix was freshly prepared just before each round of IMC staining in order to minimise the antibody mixing time. Moreover, we did not see markers that from different major cell groups mixing. The validations of each antibody we performed by comparing IF and IMC signal described above showed no evidence of metals diffusion or interaction between antibodies.

- Fig 4: The second part of Fig4 (f-h) has been removed. Please explain why. The intensity scale in 4c does not allow to identify whether there are changes in intraAb for cluster 8 and cluster 32 neurons in between groups. Please adjust or present boxplots for those.

In the previous version of the paper, we included the second part of Fig.4 (panels f to h) showing the percentage of cells per clusters simultaneously positive for both intraA β and pTau, their cortical location and examples of IMC immunostaining with intraA β ⁺pTau⁺RORB⁺ cells. The aim of this analysis was to determine whether neurons accumulating either intraA β or pTau would progressively transition into accumulating the other pathological protein too. However, as cluster_10 RORB+GPC5+ neurons are the only population with pTau accumulation, they also appear as the only population being intraA β ⁺pTau⁺. Moreover, the transition between accumulation of pathological protein based exclusively on these results would have been difficult to prove. We concluded that these results were difficult to interpret and were not adding any significant information not already present in previous panels. We therefore decided to exclude them from the new version of the manuscript to avoid confusion.

There are no meaningful differences in percentages of intraA β ⁺ cells between groups in cluster 8 (CtrlCV: 99.87 \pm 0.19%; CtrlTREM2: 99.85 \pm 0.37%; AlzCV: 99.42 \pm 1.27%; AlzTREM2: 98.26 \pm 2.83%) and cluster 32 (CtrlCV: 100 \pm 0%; CtrlTREM2: 100 \pm 0%; AlzCV: 99.31 \pm 2.95%; AlzTREM2: 100 \pm 0%). Raw numbers visualised in these plots are available in the source data provided.

- Fig 4: The authors define intracellular AT8 pos Tau as NFT. Please provide some structural means (e.g. ThT or and PHF) to validate that this is fibrillar tau.

We agree with the reviewer that indeed we have no data that directly establishes that the ptau identified is fibrillar. All NFT references in the manuscript therefore have been changed to pTau.

- Fig S5: The authors show that Moab2 stains intracellular Abeta. The study would benefit from some additional stainings/images using this approach to validate the IMC findings such as for e.g. the GAD1+ or RORB+ neurons, demonstrating changes in intracellular Abeta across different groups/Braak stages.

We have performed further immunofluorescences staining using MOAB2/GAD1 or MOAB2/RORB in each sample group (CtrlCV, CtrlTREM2, AlzCV, AlzTREM2) to show accumulation of intraA β in both neuronal populations across Braak stages. The images were added as Fig.S6.

- Fig 5: The 3D reconstruction is difficult to discern and should be increased.

The 3D reconstruction in Fig.5 has now been made larger and should be easier to view.

- Fig 5/Line 402: The authors interpret their findings of CD68/GAD1 double positivity that those microglia phagocytose GAD1+ neurons. Again, some high res imaging results showing how activated microglia phagocytose those neurons would enhance this part of the study.

We agree with the reviewer that the immunofluorescence staining does not provide sufficient data to unequivocally demonstrate that microglia can

phagocytose GAD1⁺ neurons. Dynamic imaging or electron microscopy would be more definitive. However, previous studies have relied on co-localisation between microglial and neuronal or synaptic markers as evidence for phagocytosis, as here (De Shepper et al., 2023; Damisah et al., 2020). Despite the independent interest, we see definitive resolution of the question as being outside the scope of the current study. We have added a sentence in the results to highly the potential uncertainty.

Results, page 16:

“These results suggest CD68⁺GAD1⁺ (cluster_15) cells, which are more abundant in TREM2 risk variant tissue sections, might define microglia that have phagocytosed GAD1⁺ inhibitory neurons or their synapses. However, additional higher resolution imaging studies (beyond the scope of this work) are needed to confirm this hypothesis.”

- Line 448: The authors show that S100⁺ astrocytes spatially interact with RORBMAP2 and GAD1ADARB1 neurons. They conclude however that astrocytes associate spatially with resilient neurons, which would be cluster 10 (RORB1GPC5).

We thank the reviewer for spotting the confusion in this sentence. Our results show that CD68⁺GAD1⁺ microglia spatially associate with 6 neuronal populations, half of which are vulnerable neurons or neurons accumulating pTau. Conversely, homeostatic S100B⁺GFAP⁻ astrocytes associate with 9 neuronal populations, only 2 of which are vulnerable neurons. Because there is a higher proportion of vulnerable neurons among the neuronal populations to which CD68⁺GAD1⁺ microglia associates to compared to S100B⁺GFAP⁻ astrocytes, we concluded that, relatively to CD68⁺GAD1⁺ microglia, S100B⁺GFAP⁻ astrocytes show a stronger association with resilient neurons. This paragraph has been changed to clarify this aspect.

Results, page 18:

“Among all analysed samples, we found that CD68⁺GAD1⁺ microglia interacted with RORB⁺GPC5⁺ (cluster_10), RORB⁺FOXP2⁺ (cluster_8) and LMO3⁺ (cluster_27) excitatory (Fig.5.d) and PVALB⁺CCK⁺ (cluster_29), CALB1⁺ (cluster_22) and GAD1⁺FOXP2⁺ (cluster_32) inhibitory neurons (Fig.5.e). Therefore, half of the neuronal clusters spatially associated with CD68⁺GAD1⁺ represent neuronal populations accumulating pathological proteins. Conversely, among the astrocytic clusters, only S100B⁺GFAP⁻ astrocytes (cluster_3, cluster_35) showed preferential spatial interactions with neuronal clusters. Of these nine neuronal clusters, only two included vulnerable neurons (RORB⁺MAP2⁺ [cluster_5] and GAD1⁺ADARB1⁺ [cluster_9]) and these did not show accumulation of pathological proteins (Fig.S7.f,g). Together, these data suggest that activated microglia may preferentially cluster around vulnerable neurons and neurons accumulating pTau, while homeostatic astrocytes have a weaker spatial association with vulnerable neurons and are associated more frequently with resilient neurons.”

- line 484: the authors refer to a preprint describing a similar snRNAseq study. please detail potential overlap.

The Fancy et al., 2022 preprint originates from our lab. The work has not been completed for full publication. In this preliminary study, the snRNAseq dataset was generated from a large human post-mortem tissue cohort including multiple brain regions and used to explore astrocytic and microglial transcription signatures and their differences between common allele and in TREM2 variant carriers. The intent of the Fancy et al., 2022 manuscript is therefore fundamentally different from that of the current manuscript. The current manuscript is the only one from our group under submission that is reporting this dataset. The result sentence introducing this study has been changed accordingly to clarify the dataset origin.

Results, page 19:

“To explore pathways and molecular mechanisms associated with vulnerability of the neuronal subtypes identified, we analysed a single nuclear RNA sequencing (snRNAseq) dataset (described in an earlier preprint from our group [Fancy et al., 2022]) generated from cryo-preserved blocks from homologous MTG regions of the same brains studied above, along with 10 additional MTG samples to enhance study power (3 CtrlCV, 1 CtrlTREM2, 1 AlzCV and 5 AlzTREM2) (Supp. Table 4).”

REVIEWERS' COMMENTS

Reviewer #2 (Remarks to the Author):

The authors have been very responsive to the issues raised in the earlier review. The revised MS has addressed all of my earlier concerns in a satisfactory manner. There is need for one correction on page 27.

Results, page 27:

“Chloride channel-related pathways were enriched at the early or mid stages of disease in both Exc-L5-RORB-LINC01202 and L6-THEMIS-LINC00343 clusters, (e.g., “chloride channel activity” in Modules 7 and 4 modules, respectively; Fig.6.g, Fig.S10.e). Chloride channels regulate neuronal excitability. Their homeostasis is altered in AD and their restoration has been suggested as a potential target for reversing cognitive decline previously.”

The modules numbers need to be corrected. They should be Modules 3 and 2, respectively instead of 7 and 4.

We thank the reviewer for spotting these mistakes. Module numbers have been corrected in that whole result section.

Reviewer #3 (Remarks to the Author):

Rev: The authors made a substantial effort to clarify the previously raised comments. Some minor aspects however should be further clarified. Please see my comments in red in line with authors responses.

- Fig 3: Previously, concerns were raised about the cell count/density statistics. The authors assured that any disagreements in between the plots were a plot error and submit revised versions of the plots as well as cell density stats across all cortical layers. What is the accuracy and precision of the IMC panel? How many technical replicates per patient (not per section) were collected? What is the %CV of the method within individuals?

The reviewer raises appropriate questions regarding the reliability of the quantitative IMC analyses. The first question is accuracy. We determined accuracy of IMC relative to an immunofluorescence (IF) “gold standard”. Neuronal antibodies added to the panel were first validated with IF relative to published examples (provided by the manufacturer) of prior uses of the antibodies and in conjunction with established pan-neuronal or synaptic markers. The quality and relative intensity of antibodies’ signals in IMC compared to this IF “gold standard” was used to qualitatively determine the reliability of the IMC immunostaining. We have added a new supplementary figure with IF and IMC images of all new neuronal antibodies that passed these quality control steps and were used in this study (Fig.S1).

Rev: While this reviewer appreciates the effort, it is hard to discern from the Figure and the legend what is what and how the IF signal compares to the IMC. I presume you aim to show a relative/qualitative assesment of staining pattern, i.e. nr of pos. staining in rel. comparison to the panneuronal marker. This should be clarified in the methods and the legend. Also, specify in the legend that the Ir191 signal is a nuclear marker and what metals were used for the respective antibodies.

Additional information has been added to the methods and legend of Figure S1.

Methods, page 26, l. 823-824:

“The quality and relative intensity of antibody signals in IMC compared to the IF gold standard was used to qualitatively determine the accuracy of the IMC immunostaining.”

Supplementary figures, page 3:

“Supplementary Figure 1. **Immunofluorescence (IF) and imaging mass cytometry (IMC) signal from 16 neuronal markers.** The shortlisted new neuronal markers (Fig.1b; Supp. Table 2) were tested by comparing their immunostaining in paired sections from the same brains with IF or IMC in conjunction with established pan-neuronal or synaptic markers (MAP2, MAP2all, NeuN or synaptophysin) and an iridium nuclear marker (Ir191). Only neuronal markers that were validated in this way are shown here (16/34 tested). These were used in the IMC panel applied for the studies described. The IMC

antibodies used in these tests were conjugated to the same metal isotopes as the finalised panel (Supp. Table 3). Yellow arrowheads indicate cells double positive for the tested and pan-neuronal marker, while red arrowheads indicate cells only positive for the tested neuronal marker. Scale bars represent 100µm.”

The reliability of our antibody panel as well as our method for detecting and classifying neuronal and glial populations also was confirmed by comparing proportions of glial populations and excitatory/inhibitory neurons expected from prior published reports vs. those observed (Fig.2c) and the relative cortical layer locations of neuronal populations in CtrlCV samples (Fig.2d).

Rev: For 2c: I am not sure that the rel. number of oligodendrocytes corresponds to the Lit value. Please clarify this deviation.

We thank the reviewer for spotting this deviation. We have added these sentences to the results and methods sections to explain the discrepancies in oligodendrocytes counts.

Results, page 5. l. 127-129:

“This discrepancy likely is due to the limited number of markers used for identifying microglia, which will bias towards overcounting these cells, and the relatively weak IMC signal from OLIG2, which may lead to undercounting of oligodendrocytes.”

Methods, page 23, l. 717-718

“The OLIG2 marker gave a relatively less strong and more variable IMC signal than for other markers.”

To allow the reviewer to better appreciate our approach, we have first, as noted in the response to Reviewer 1 above, made all raw and processed IMC images a SIMPLI output files are now available for reviewers to inspect (<https://figshare.com/s/311a85639e9de2a4e502>). R scripts used for data analysis of SIMPLI output files are available in a GitHub repository (<https://github.com/AlessiaCaramello/Vulnerable-neurons-in-AD>). All data presented in this study thus can be re-generated using our R scripts using either SIMPLI output files or raw IMC images (which will require processing with SIMPLI). Our results are based on the means of values from 3 ROIs per patient, all were selected from the same section. We expanded the methods section to clarify how ROIs were selected and acquired as described below. Methods, page 33: “One FFPE section per sample was stained with the full conjugated antibody panel (Supp. Table 3) following the same protocol as for immunofluorescence. After overnight incubation with metal-tagged antibodies and an iridium (Ir191/193) intercalator (Standard BioTools, CA, USA) to identify nuclei, slides were washed with water 3 x 10 minutes each and air dried for 20 minutes. IMC was performed using a Hyperion Tissue Imager coupled to a Helios mass cytometer (Fluidigm, Standard BioTools, CA, USA). The instrument was first tuned using the manufacturer’s 3-Element Full Coverage Tuning Slide before the slides were loaded into the device. Three ROIs were ablated from the same FFPE section for each sample at a laser frequency of 200Hz with 1µm resolution. Each ROI was spanning the full

cortical depth from L1 to L6 (grey matter – representative ROI in Fig.1.B) adding up to a total of 6 hours of acquisition per sample (~2h per ROI, corresponding to 1.2 ± 0.15 mm² of area ablated per ROI) keeping the total area acquired per sample equal (total 3.62 ± 0.11 mm² ablated area per sample). Data was stored as .txt files that were used for subsequent processing (Fig.1.d).”

To estimate the precision of our IMC antibody panel in detecting neuronal populations, we performed a new IMC experiment with the antibody panel used to generate the primary data in the manuscript, acquiring 3 ROIs from 3 different sections of MTG from the same patient (CtrlTREM2 – NP0122013_1, NP0122013_2, NP0122013_3). Images were processed with SIMPLI using the same settings as in our original study and detected nuclei were clustered (resolution, 1.2). 24 clusters were identified (Revision Figure 2.a), with evenly distributed contributions from all of the slides/technical replicates (Revision Figure 2.b). Marker expression in each cluster (Revision Figure 2.c) was used for assigning clusters to cell types (Revision Figure 2.d), as described in the manuscript.

The pattern of clustering was not identical to that in reported in the manuscript. This is due to the lower number of input cells which limited Seurat clustering ability despite using a higher resolution (1.2 here, compared to 0.9 in the original study). In this new, smaller dataset, we identified 5 clusters of excitatory neurons, 6 of inhibitory neurons, 3 of unclassified neurons, 2 of astrocytes, 2 of microglia and 3 of oligodendrocytes. All glial and neuronal populations identified showed a direct correspondence with those found in our original study UMAP, e.g., the RORB+FOXP2+ cluster_18 (similar to cluster_8 in the original study) and CD68+Iba1+GAD1+ cluster_19 (similar to cluster_15 in the original study). As fewer clusters were identified in this analysis of a small dataset compared to that in the original study (23 vs. 36), 13 clusters identified in the original study do not show a direct correspondence to clusters in this new analysis (4x excitatory neurons, 6x inhibitory neurons, 1x astrocytes and 1x oligodendrocytes). The markers of neuronal populations missing as distinct clusters in this new analysis are present in the two “positive for all” clusters (cluster_1 and cluster_23), suggesting that data from a larger number of cells might have allowed these populations to be clustered separately. Revision Figure 2 – Clustering and cell type assignment of cells identified from 3 ROIs from 3 slides from the same sample. (a,b) UMAP plots of clusters identified in this analysis, shown by cluster identity (a) and slide/ROI origin (b) (NP0122013_1 ROIs #2,3,4, NP0122013_2 ROIs #1,2,3, NP0122013_3 ROIs #2,3,4). (c) Heatmap of intensities of IMC marker expression in each cluster and preliminary assignments to cell types. (d) Final assignment of clusters to neuronal and glial populations based on markers expression shown in (c).

We then computed the coefficient of variation (CV) of the numbers of cells in cluster generated from 3 individual ROIs in single sections from the same individuals (Revision Figure 3a). The average CV in CtrlCV samples was $30.06 \pm 11.56\%$ for all clusters and $29.7 \pm 10.9\%$ for neuronal clusters only, with a trend towards higher CV in smaller clusters (cluster_31 to cluster_35).

However, the precision for our study was higher than this. We acquired multiple ROIs per sample to minimise the impact of variation from ROI to ROI in the same section and better estimate the true values as the means across the comparable ROIs. The

CV between the means across 3 ROIs from 3x technical replicates decreased to $15.3\pm 7.6\%$ for all clusters and $13.73\pm 8.32\%$ for neuronal clusters only (Revision Figure 3b).

Revision Figure 3 – Coefficient of variation (CV) between ROIs from the section individual and between technical replicates of means across three ROIs of three sections taken from the same individual. (a) CV of cell number per cluster between ROIs of the same individual from our original study dataset, grouped by disease (Ctrl/Alz) and TREM2 variants (CV/TREM2). (b) CV of mean cell number per cluster between technical replicates of three slides from each of three different control brains (CtrlTREM2 – NP0122013_1, NP0122013_2, NP0122013_3).

Rev: This effort is highly appreciated and the Figures should go in the SI.

We appreciate the reviewer comment regarding the new analysis we performed. The figures and analyses generated for the revision have been added to the final manuscript version in the methods section and as a supplementary figure (Fig.S4).

Methods, page 26, l. 822-841:

“Evaluation of accuracy and precision of IMC antibody panel

The quality and relative intensity of antibody signals in IMC compared to the IF gold standard was used to qualitatively determine the accuracy of the IMC immunostaining. To estimate the precision of our IMC antibody panel in detecting neurons, we performed an independent IMC experiment with the full the antibody panel, acquiring 3 ROIs from three different sections of MTG from the same CtrlTREM2 brain. Images were processed with SIMPLI using parameters consistent with those used throughout and detected nuclei were clustered at 1.2 resolution. 24 clusters were identified (Fig.S4.a). The clusters showed similar contributions from each of the technical replicates (Fig.S4.b). Marker expression per cluster was inspected (Fig.S4.c) and used for assigning clusters to cell types (Fig.S4.d). The coefficient of variation (CV) of the number of cells per cluster was calculated as the standard deviation of dataset divided by the mean of dataset⁸⁸ (Fig.S4.e,f).

The average coefficient of variation (CV) across ROIs for the neuronal clusters was $29.70\pm 10.90\%$ ($30.06\pm 11.56\%$ across all clusters) (Fig.S4.e). The CV between averaged ROIs from three technical replicates in serial sections was $13.73\pm 8.32\%$ for neuronal clusters ($15.30\pm 7.60\%$ for all of the clusters together) (Fig.S4.f). Power estimates based on the CV for cluster_8 (24.4%) suggested that 8-9 samples were needed to detect differences between sample groups.“

- Some concern remains regarding the observed differences which appear as stat. significant in one analysis but lost for some densities when including more groups, while not affected for others. Eg cluster 10 (RORB/GPC5) neurons are higher in L6 in Ad as compared to Ctrl (Fig 3b) though this significance is lost in Fig 3d (L6) upon inclusion of CtrlTrem2 and ADTrem2. Conversely, significance is maintained for difference in cluster 32 neurons (GAD1/FOXP2) (L3+L6, Fig 3b) vs (L3+L6, Fig 3d).

We understand the reviewer uncertainty. The differences in p values for the cluster contrasts described by the reviewer appear to change because the different statistical analyses were applied. When comparing two groups (CtrlCV and AlzCV) we applied Wilcoxon signed-rank test, while when comparing four groups (CtrlCV, AlzCV, CtrlTREM2, AlzTREM2) we applied ANOVA and Tukey tests or Kruskal–Wallis and Wilcoxon signed-rank test depending on whether groups showed normal or non-normal distributions, respectively. When more than two groups were tested, p values of pair-group comparisons (from Tukey test after ANOVA test and Wilcoxon signed-rank test after Kruskal–Wallis test) were adjusted for multiple comparisons using the FDR method. For example, the p value for numbers of L6 neurons in cluster 10 (RORB/GPC5) for the CtrlCV-AlzCV comparison was 0.033, but a p value of 0.214 was found for the CtrlCV-AlzCV-CtrlTREM2-AlzTREM2 ANOVA. Similarly, the p values for cluster 32 (GAD1/FOXP2) neurons in L3 and 6 for the CtrlCV-AlzCV comparison were 0.003 and 0.005, respectively, but were 0.018 and 0.025, respectively, for the CtrlCV-AlzCV-CtrlTREM2-AlzTREM2 comparison. We have updated the material and methods section to better clarify our statistical analyses as below.

Material and methods, page 37:

Rev: This reviewer is familiar with the basics using different stat. tests such as that multiple group comparison requires multiple testing and an adjusted p-values. I suggest, the authors should rather clarify that the trends are maintained despite loss of stat. significance due to multiple testing.

We thank the reviewer for this suggestion. A sentence explaining the apparent loss of statistical significance has been added to the relevant result section.

Results, page 7, l. 194-196:

“The previously identified differences between CtrlCV and AlzCV samples in layer-specific neuronal densities (Fig.3.b) still showed the same trend despite loss of statistical significance with multiple testing corrections.”

- Fig S5: The authors show that Moab2 stains intracellular Abeta. The study would benefit from some additional stainings/images using this approach to validate the IMC findings such as for e.g. the GAD1+ or RORB+ neurons, demonstrating changes in intracellular Abeta across different groups/Braak stages.

We have performed further immunofluorescences staining using MOAB2/GAD1 or MOAB2/RORB in each sample group (CtrlCV, CtrlTREM2, AlzCV, AlzTREM2) to show accumulation of intraA β in both neuronal populations across Braak stages. The images were added as Fig.S6.

Rev: These images are very much appreciated. Fig S6 does not convey any relation of intracellular Abeta to Braak stages. Please specify in the text and the legend what Braak stage the images in S6 refer to. Do you have similar images for all groups? Otherwise state that these are just representative as proof of principle.

Braak stages of samples shown in Fig.S7 have been added both in the figure, figure legend and main text.

Results, page 10, l 300-303:

“We found that GAD1⁺ and RORB⁺ neurons co-localised with MOAB-2 immunostaining (representative images from CtrlICV [Braak 2], CtrlTREM2 [Braak 0], AlzCV [Braak 6] and AlzTREM2 [Braak 6] samples shown in Fig.S7.a,d) which was found intracellularly (Fig.S7.b,c,e,f).”

Minor comments:

- The SI Table pdf only contains Suppl Tab 5
- Legend Fig S2. Should say “ROI”

These were corrected.